# Constraint-Aware Discrete Black-Box Optimization Using Tensor Decomposition

## Abstract

Discrete black-box optimization has been addressed using approaches such as Sequential Model-Based Optimization (SMBO), which aim to improve sample efficiency by fitting surrogate models that approximate a costly objective function over a discrete search space. In many real-world problems, the set of feasible inputs, such as valid parameter configurations in engineering design, is often known in advance. However, existing surrogate modeling techniques generally fail to capture feasibility constraints associated with such inputs. In this paper, we propose a surrogate modeling approach based on tensor decomposition that captures the structure of discrete search spaces while directly integrating feasibility information. To implement this approach, we formulate surrogate model training as a constrained polynomial optimization problem and solve a relaxed version of it. Our experiments on both synthetic and real-world benchmarks, including a pressure vessel design task, demonstrate that the proposed method improves sample efficiency by effectively guiding the search away from infeasible regions.

## 1 Introduction

Black-box optimization (BBO) aims to find optimal inputs for an objective function that can only be accessed through input-output data (Rios & Sahinidis, 2013a) and has been widely used in fields like engineering design (Coello & Montes, 2002), material discovery (Frazier & Wang, 2016), and hyperparameter tuning for machine learning (He et al., 2021; Bergstra et al., 2011). Since evaluating such objective functions is often costly in terms of monetary cost, execution time, and computational resources, sample-efficient methods like Sequential Model-Based Optimization (SMBO) have been developed (Hutter et al., 2011; Shahriari et al., 2015). SMBO uses a surrogate model to approximate the objective function and an acquisition function to balance exploration and exploitation when choosing new samples.

This paper focuses on discrete search spaces that are commonly encountered in real-world applications, such as categorical parameters representing the choices of specific components in engineering design (Papalexopoulos et al., 2022; González-Duque et al., 2024; Zamuda et al., 2018). For such discrete BBO problems, methods based on tensor decomposition (TD) offer a sample-efficient approach that has recently demonstrated great potential (Sozykin et al., 2022; Chertkov et al., 2022; Batsheva et al., 2023). In this approach, a discrete search space is represented by a tensor, and a surrogate model is constructed by approximating this tensor, for example, with a low-rank tensor.

For many real-world problems addressed by BBO, consideration of input constraints arising from safety requirements, manufacturing capabilities, or design rules is crucial. A typical approach to introduce these input constraints into SMBO is to evaluate the feasibility at the stage of acquiring new samples, for example by rejecting infeasible inputs or modifying the acquisition function (Gardner et al., 2014; Gelbart et al., 2014). More advanced methods also follow this paradigm, employing sophisticated solvers to optimize the acquisition function over the known feasible domain (Papalexopoulos et al., 2022). In these approaches, however, the surrogate model itself is typically learned without considering feasibility, which can reduce sample efficiency, especially when the feasible domain is small. To address this limitation, introducing *constraint-awareness* into the surrogate model, i.e., considering feasibility during training the surrogate model, is expected to enhance the approximation of the objective function within the feasible region and increase the sample efficiency of SMBO.

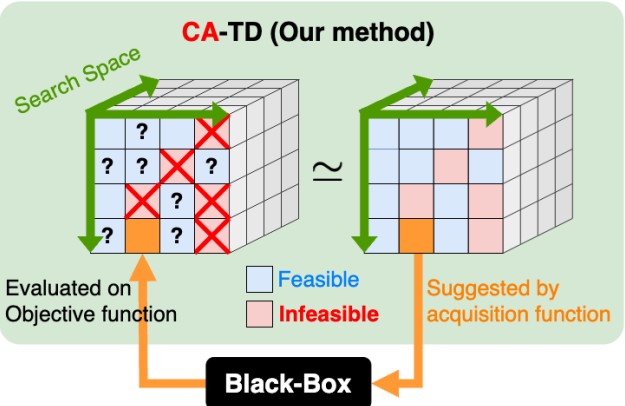

Figure 1: Overview of CA-TD: CA-TD approximates the black-box objective over feasible inputs, effectively guiding the search in SMBO.

In this paper, we address a BBO problem in discrete space with input constraints, where the objective function is expensive to evaluate, and the input constraints are known as a prior, i.e., cheap to evaluate constraints. To address this problem, we propose a TD-based surrogate model using the Tensor-Train (TT) decomposition to deal with discrete space under input constraints, named Constraint-Aware Tensor Decomposition (CA-TD) (Figure 1). In our approach, we formulate the task of learning a surrogate model under input constraints (surrogate learning) as a constrained polynomial optimization problem (POP). To efficiently solve this POP, we introduce a scalable approximation by incorporating constraint violations into the loss function for surrogate learning as a differentiable penalty term, which enables efficient gradient-based training.

Our contribution is threefold:

- We formulate the training of a TD-based surrogate model with input constraints as a POP, which defines our method CA-TD
- We develop a gradient-based training method for CA-TD by introducing a penalty term for constraint violations for scalability
- We evaluate our approach on a diverse set of synthetic and real-world benchmarks, including a classic engineering design task. The results demonstrate improvements in sample efficiency compared to conventional methods.

Our source code and datasets are publicly available at `https://github.com/xxxxx`.

## 2 PRELIMINARIES

This section introduces the fundamental concepts underlying our proposed method. Since our problem formulation is based on the discrete BBO approach within the SMBO framework, we begin by providing an overview of BBO and SMBO. Additionally, we describe TD-based surrogate models that our method employs, focusing in particular on tensor train (TT) decomposition.

### 2.1 DISCRETE BLACK-BOX OPTIMIZATION PROBLEM AND SEQUENTIAL MODEL-BASED OPTIMIZATION

First, we formulate the discrete BBO problem that is the basis of the problem addressed in this paper. Given a search space $\mathcal{X} = X_1 \times \cdots \times X_d$, where $X_k$ is a finite set for $k = 1, \ldots, d$, and an objective function $g : \mathcal{X} \to \mathbb{R}$. The goal of this problem is to find

$$\mathbf{x}^\star = \arg\min_{\mathbf{x} \in \mathcal{X}} g(\mathbf{x}).$$

In this problem, no further information about $g$ is available, such as its derivative, so it is called a black-box function. In practice, it is assumed that evaluating $g$ is costly, and it is desirable to obtain a good solution with as few evaluations of $g$ as possible.

---

**Algorithm 1** The Procedure of SMBO

---

**Require:** Objective function $g$, search space $\mathcal{X}$, surrogate model class $f$, acquisition function $\alpha$, maximum iterations $T$.

1: Initialize history $\mathcal{H}$
2: **for** $t = 1$ to $T$ **do**
3:   Fit or update the surrogate model $f_{t-1}$ using $\mathcal{H}$.
4:   Select the next point to evaluate:
5:     $\mathbf{x}_t \leftarrow \arg\max_{\mathbf{x} \in \mathcal{X}} \alpha(\mathbf{x}, f_{t-1})$.
6:   Evaluate the objective function: $y_t \leftarrow g(\mathbf{x}_t)$.
7:   Augment the history: $\mathcal{H} \leftarrow \mathcal{H} \cup \{(\mathbf{x}_t, y_t)\}$.
8: **end for**
9: **return** $\mathcal{H}$

---

The overall SMBO procedure is summarized in Algorithm 1. SMBO is a general framework of BBO that includes Bayesian optimization as a special case, and we adopt a variant widely used in Bayesian optimization (Frazier, 2018; Shahriari et al., 2015; Bergstra et al., 2011), which iteratively performs the following three steps. 1) A probabilistic surrogate model $f$ is fitted to all previous observations. Instead of directly evaluating the costly function $g$, the surrogate model $f$ is used to approximate $g$. 2) The most promising point to evaluate next is selected by using an acquisition function $\alpha(\cdot)$, defined based on the surrogate function $f$. In our implementation, we utilize the Expected Improvement (EI) criterion (Mockus et al., 1978) as the acquisition function to decide the next point. 3) The objective function $g$ at the selected point is evaluated. This loop is repeated $T$ times.

### 2.2 Tensor-Train Surrogate Model

This section briefly describes the method to use TT decomposition as a TD-based surrogate model $f$ for approximating black-box functions $g$. The TT decomposition provides a compact representation of high-dimensional tensors by factorizing them into a sequence of smaller core tensors.

Before considering the surrogate model, we first consider the tensor for storing the evaluated points $\mathcal{Y} \in (\mathbb{R} \cup \emptyset)^{|X_1| \times \cdots \times |X_d|}$, where $|X_k|$ is a cardinality of a finite set $X_k$ and $\emptyset$ represents a point that have not yet been evaluated. Note that this tensor has the same size as the discrete search space $\mathcal{X}$. By denoting the element of $\mathcal{Y}$ corresponding to point $\mathbf{x} = (x_1, \ldots, x_d) \in \mathcal{X}$ as $\mathcal{Y}[\mathbf{x}]$, each entry $\mathcal{Y}[\mathbf{x}]$ represents the value of the objective function $g(\mathbf{x})$. In the context of SMBO, the values of $g(\mathbf{x})$ that have already been evaluated are stored in $\mathcal{Y}[\mathbf{x}]$.

Here, a TD-based surrogate model is provided by using TT decomposition to approximate the above ground-truth tensor $\mathcal{Y}$ to a low-rank $\hat{\mathcal{Y}}$, which can be expressed as follows:

$$\hat{\mathcal{Y}}[\mathbf{x}] = \mathbf{G}^{(1)}[x_1] \mathbf{G}^{(2)}[x_2] \cdots \mathbf{G}^{(d)}[x_d], \tag{1}$$

where $\mathbf{G}^{(k)}[x_k] \in \mathbb{R}^{r_{k-1} \times r_k}$ denotes the $x_k$-th lateral slice of the $k$-th core tensor ($k = 1, 2, \ldots, d$). Note that $r_k \in \mathbb{N}$ ($r_0 = r_d = 1$), referred to as the TT-ranks, controls the expressiveness of the decomposition. The number of parameters scales as $\mathcal{O}(d\,n\,r^2)$, where $n = \max_k |X_k|$ and $r = \max_k r_k$. Instead of evaluating the objective function $g$, a low-rank surrogate tensor $\hat{\mathcal{Y}}$ that imputes the unobserved elements of $\mathcal{Y}$ can be used as a surrogate model $f(\mathbf{x}) = \hat{\mathcal{Y}}[\mathbf{x}]$ by controlling the TT-ranks $r_k$.

In TT decomposition, a surrogate model is trained to minimize the mean squared error $\mathcal{L}_{\text{recon}}$ between the surrogate model output $\hat{\mathcal{Y}}[\mathbf{x}]$ and the observations $\mathcal{Y}[\mathbf{x}]$ in the previously evaluated dataset $H = \{\mathbf{x} : \mathcal{Y}[\mathbf{x}] \neq \emptyset\}$:

$$\mathcal{L}_{\text{recon}} = \frac{1}{|H|} \sum_{\mathbf{x} \in H} \left( \mathcal{Y}[\mathbf{x}] - \hat{\mathcal{Y}}[\mathbf{x}] \right)^2. \tag{2}$$

## 3 PROPOSED METHOD: CONSTRAINT-AWARE TENSOR DECOMPOSITION SURROGATE

Our method integrates constraint-awareness directly into the learning process of the TD-based surrogate model for SMBO.

### 3.1 PROBLEM FORMULATION

First, we describe our formulation of the discrete BBO under input constraints. Given a discrete search space $\mathcal{X} = X_1 \times \cdots \times X_d$ and an objective function $g : \mathcal{X} \to \mathbb{R}$, the goal is to find

$$\mathbf{x}^\star = \arg\min_{\mathbf{x} \in \mathcal{X}} g(\mathbf{x}) \quad \text{subject to} \quad c(\mathbf{x}) = 1, \tag{3}$$

where a constraint function $c : \mathcal{X} \to \{0, 1\}$ whose evaluation cost is negligible compared to $g$.

For ease of handling, we introduce the notation $\mathcal{X}_{\text{feas}} := \{\mathbf{x} \in \mathcal{X} \mid c(\mathbf{x}) = 1\}$ and rewrite the above problem as:

$$\mathbf{x}^\star = \arg\min_{\mathbf{x} \in \mathcal{X}_{\text{feas}}} g(\mathbf{x}).$$

### 3.2 A FORMULATION AS A POLYNOMIAL OPTIMIZATION PROBLEM

We incorporate input constraints into the surrogate model by assuming that evaluations at infeasible inputs yield objective values greater than or equal to a threshold $\tau$, since we solve a minimization problem. Specifically, we define the infeasible subset of the search space as $\mathcal{X}_{\text{infeas}} := \mathcal{X} \setminus \mathcal{X}_{\text{feas}}$ and impose the condition

$$g(\mathbf{x}) \geq \tau \quad \text{for all} \quad \mathbf{x} \in \mathcal{X}_{\text{infeas}},$$

where the threshold $\tau$ is set to the maximum objective value observed so far among feasible inputs. Thus, the surrogate model learning under the input constraints is formulated as follows:

$$\min_{\hat{\mathcal{Y}}} \quad \frac{1}{|H|} \sum_{\mathbf{x} \in H} \left( \mathcal{Y}[\mathbf{x}] - \hat{\mathcal{Y}}[\mathbf{x}] \right)^2, \tag{4}$$

$$\text{subject to} \quad \hat{\mathcal{Y}}[\mathbf{x}] \geq \tau \quad \text{for all} \quad \mathbf{x} \in \mathcal{X}_{\text{infeas}}.$$

Since the surrogate tensor $\hat{\mathcal{Y}}[\mathbf{x}]$ is a polynomial function of the core tensor parameters according to Equation 1, the above problem constitutes a POP (Appendix G). An established approach to solving POPs involves constructing a hierarchy of semidefinite programming (SDP) relaxations (Lasserre, 2001), which can provide arbitrarily tight lower bounds on the global optimum. We employ this approach to the problem Equation 4 and simply refer to it as HSDP hereafter.

### 3.3 PENALIZED LOSS FUNCTION

Since POPs are NP-hard and HSDP remains computationally demanding, we relax the hard constraints by incorporating a differentiable penalty term into the objective. Specifically, the constrained optimization problem in Equation 3 is relaxed to an unconstrained optimization:

$$\arg\min_{\mathbf{x} \in \mathcal{X}} \quad g(\mathbf{x}) + \lambda\, h(c(\mathbf{x})),$$

where $h$ is a penalty function that measures the violation of the constraint $c(\mathbf{x}) = 1$ and $\lambda > 0$ controls the trade-off between data fitting and constraint enforcement.

To solve this relaxed problem while promoting satisfaction of the constraint $c$, we train the TT-based surrogate model by minimizing the following total loss using gradient-based optimization:

$$\mathcal{L}_{\text{total}} = \mathcal{L}_{\text{recon}} + \lambda \mathcal{L}_{\text{pen}}$$

$$= \frac{1}{|H|} \sum_{\mathbf{x} \in H} \left( \mathcal{Y}[\mathbf{x}] - \hat{\mathcal{Y}}[\mathbf{x}] \right)^2 + \lambda \cdot \frac{1}{|\mathcal{X}_{\text{infeas}}|} \sum_{\mathbf{x} \in \mathcal{X}_{\text{infeas}}} \max\left( 0, \tau - \hat{\mathcal{Y}}[\mathbf{x}] \right) \tag{5}$$

The penalty term $\mathcal{L}_{\text{pen}}$ is designed to impose the constraint, aiming to push the outputs of the surrogate at least above the threshold $\tau$ at infeasible inputs, while maintaining approximation accuracy on the observed data. We denote this penalty-based strategy for the CA-TD surrogate learning as PGRAD (Penalty with Gradient-based optimization) throughout the rest of the paper.

### 3.4 Uncertainty Quantification using Ensembles for Acquisition Function

In SMBO, acquisition functions determine which input to evaluate next by leveraging approximations from surrogate models. Their strategies are typically based on the idea of the exploration–exploitation trade-off, requiring not only an accurate approximation of the objective's value but also a reliable quantification of the uncertainty of the approximation.

To this end, we use an ensemble of $M$ independently learned TT-based surrogate models $\{\hat{\mathcal{Y}}^{(m)}\}_{m=1}^M$, each initialized with different random TT core parameters. At any input $\mathbf{x}$, the ensemble predictions $\{\hat{\mathcal{Y}}^{(m)}[\mathbf{x}]\}_{m=1}^M$ induce an empirical distribution. We compute the sample mean $\mu(\mathbf{x})$ and standard deviation $\sigma(\mathbf{x})$ from this ensemble to evaluate the EI acquisition function, whose formula is given by $\alpha_{\text{EI}}(\mathbf{x}) = \mathbb{E}\left[\max(0, y^\star - Y(\mathbf{x}))\right]$, where $y^\star = \min_{\mathbf{x} \in \mathcal{H} \cap \mathcal{X}_{\text{feas}}} g(\mathbf{x})$ is the best (minimum) feasible objective value observed so far, and $Y(\mathbf{x})$ is the predictive distribution at $\mathbf{x}$. When a new point is sampled and evaluated, the threshold $\tau$ is updated with the maximum feasible value so far.

## 4 Related Work

Our method builds on three areas of work: constrained BBO, tensor decomposition for black-box optimization (TD-BBO), and constrained tensor decomposition. We briefly review each area below and explain our unique contributions in relation to each field.

### 4.1 Constrained Black-Box Optimization

Constrained BBO deals with expensive objectives and expensive/inexpensive input constraints whose analytic forms are unknown (Rios & Sahinidis, 2013b). Most existing algorithms extend Bayesian Optimization (BO) (Frazier, 2018; Shahriari et al., 2015), which is a form of SMBO that typically uses Gaussian Processes (GPs) as surrogate models (Williams & Rasmussen, 2006). Early GP-based approaches fit separate GPs to model each constraint and incorporate the estimated feasibility into the acquisition function, typically by combining them with EI (Gardner et al., 2014; Gelbart et al., 2014). Subsequent work further extended this strategy using augmented Lagrangian methods (Picheny et al., 2016) and level-set estimation techniques (Zhang et al., 2023).

Unlike the methods mentioned above, in the case of explicit hard constraints, some approaches maximize the acquisition function within the feasible region. For example, a method that combines GPs with mixed-integer programming to maximize the acquisition function under known constraints has been proposed (Thebelt et al., 2022). A more advanced method, NN+MILP, uses piecewise-linear neural networks with acquisition maximization via mixed-integer linear programming (Papalexopoulos et al., 2022), allowing flexible integration of combinatorial constraints in discrete search spaces.

Most existing methods in the framework of constrained BBO handle constraints by modifying the acquisition function, as mentioned above. Our proposed approach is distinguished by the direct incorporation of known feasibility information into the surrogate model training process. This is expected to enforce the surrogate model itself to learn the feasibility information, aiming for a more accurate approximation of the objective function.

### 4.2 Tensor Decomposition for BBO

TD compactly represents multi-dimensional arrays and is well-suited for capturing discrete structures. OptimaTT (Chertkov et al., 2022) adopts the TT format for discrete, unconstrained BBO, while PROTES (Batsheva et al., 2023) incorporates input constraints by encoding the input feasibility as a binary tensor and using its TT decomposition to guide surrogate initialization. Although

TTOpt (Sozykin et al., 2022) also uses the TT format, it is primarily designed for continuous optimization.

In the context of TD-BBO, methods such as OptimaTT and TTOpt are primarily designed for unconstrained optimization. A pioneering approach that incorporates input constraints is the PROTES method mentioned above, which uses these to guide the initial exploration in the proxy initialization step. In contrast, focusing on the entire search process rather than just the initialization step, CA-TD incorporates the constraints into the update of the surrogate model at each optimization step. This aims to improve sample efficiency by constantly recognizing feasible regions throughout the entire search process.

### 4.3 Tensor Decomposition under Constraints

Constraint-aware tensor decomposition has primarily been studied in the context of data analysis, where domain-specific structure is imposed on factor matrices to improve interpretability or incorporate prior knowledge. For example, constraints such as non-negativity (Alexandrov et al., 2022; Yu et al., 2022), orthogonality (Halaseh et al., 2022), and smoothness via basis function expansions (Imaizumi & Hayashi, 2017) have been explored.

While previous work on constrained tensor decomposition has primarily focused on such well-behaved linear algebraic constraints, our study departs from this trend by directly imposing constraints on individual elements of the output tensor. These constraints are not based on assumptions about the latent factors of the tensor but directly reflect the feasibility conditions of the input required in the outer-loop optimization problem, representing a new application of constraints in tensor decomposition in BBO.

## 5 Experiments

We conduct two experiments to evaluate the effectiveness of CA-TD in constrained black-box optimization on discrete domains. The objectives of these experiments are: (1) to compare the performance and scalability of our proposed constrained training strategies, HSDP and PGRAD; and (2) to evaluate the effectiveness of our constraint-aware approach using tensor decomposition (CA-TD) against conventional baseline methods, including naive extensions for handling constraints.

### 5.1 Benchmarks

This subsection briefly introduces the benchmark problems used to evaluate our method. Detailed mathematical formulations for all problems are provided in Appendix B.

**Ackley** We use the standard Ackley synthetic function (Adorio & Diliman, 2005), to which we apply a simple geometric constraint boundary. The search space is an integer grid of $\{-\ell, \dots, \ell\}^2$, and the feasible region is defined by the circular constraint $x_1^2 + x_2^2 \leq r^2$. We use this problem to evaluate performance across different scales, with the specific settings for Experiment 1 and 2 detailed in Table 1.

**Pressure Vessel** This is a classic mixed-variable engineering design problem where the goal is to minimize manufacturing cost under physical constraints (Coello & Montes, 2002). We adapt this problem to our discrete setting by discretizing the two continuous variables into 10 uniform levels.

**Warcraft** This benchmark is a grid-based path optimization problem (Ahmed et al., 2022), where the goal is to find an optimal path on a map with combinatorial constraints defining path validity. We evaluate this problem on two different map sizes: $2 \times 2$ grid with $7^4$ candidate paths, and $2 \times 3$ grid $7^6$ with candidate paths.

**Diabetes** This is a real-world inspired task where constraints are derived from domain knowledge to find actionable and medically plausible treatment plans from patient data (Smith et al., 1988).

### 5.2 Experimental Setup

Each run uses a fixed evaluation (see Table 1), initialized from a random feasible input. The number of ensembles to compute the uncertainty for TD-based surrogate models is set as $M = 10$. In

Table 1: The settings of Ackley on the grid used in experiments. $\ell$ determines the grid range $\{-\ell, \ldots, \ell\}^2$, $r$ is the radius for the circular constraint, and $T$ is the number of evaluations in SMBO.

| Experiment | Grid size | $\ell$ | $r$ | $T$ |
| --- | --- | --- | --- | --- |
| | $3 \times 3$ | 1 | 1 | 5 |
| Experiment 1 | $5 \times 5$ | 2 | 2 | 15 |
| | $7 \times 7$ | 3 | 3 | 25 |
| Experiment 2 | $65 \times 65$ | 32 | 10 | 500 |

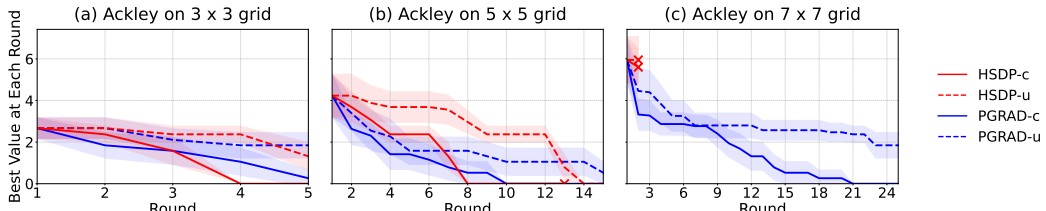

Figure 2: Optimization performance of CA-TD surrogates trained with HSDP and PGRAD on discrete Ackley benchmarks. Lower and earlier curves indicate better sample efficiency, and narrower shaded areas reflect more stable performance across runs. Solid lines denote models trained with explicit feasibility integration, while dashed lines show unconstrained variants. PGRAD (blue) offers better scalability and lower computational overhead, while HSDP (red) yields more stable convergence on small grids.

Experiment 1, the surrogate mean is used for the acquisition function for simplicity, and EI is applied in Experiment 2. The metrics include the best value obtained from the objective function and the round in which this value first appeared. All results are averaged over 10 seeds.

HSDP is implemented using the Ncpol2sdpa (Wittek, 2015) package with relaxation order 2. The generated SDP problems are solved using a sparse semidefinite programming solver.

PGRAD uses the Adam optimizer (Kingma & Ba, 2014) to minimize the total loss (Equation 5). The surrogate tensor is normalized to the range $[0, 1]$, and the penalty coefficient is fixed at $\lambda = 1$. For ablation studies with varying values of $\lambda$ in Appendix D.1. Training continues across SMBO rounds without reinitializing the tensor cores. This minimization at each round is terminated either when the loss drops below 0.1 or after 1000 epochs, whichever occurs first. The detailed implementation, including software and hardware, is provided in Appendix C.

For comparison, we consider four baseline methods, each with an unconstrained (-u) and a constrained (-c) variant. As typical BBO methods, we use Bayesian optimization based on Gaussian process (GP-u) and the Tree-structured Parzen Estimator (TPE-u) (Watanabe, 2023). The naive constrained variants (GP-c, TPE-c) are informed of the feasible space by training them offline on 200 randomly sampled infeasible inputs with a penalty value assigned. In these naive methods, if an infeasible point is selected, it is assigned the worst possible evaluation value for each task. The ablation study related to this sample size of offline-trained infeasible points is described in Appendix D.2. Also, as a conventional method for constrained tensor-based BBO, PROTES-c is included in the comparison (Batsheva et al., 2023). Furthermore, we include an advanced method (NN+MILP-c) (Papalexopoulos et al., 2022), which uses a piecewise-linear neural network as a surrogate and handles constraints via mixed-integer linear programming (MILP). For a comprehensive comparison, we also include the comparison methods run without task-specific constraints (NN+MILP-u / PROTES-u).

### 5.3 EXPERIMENT 1: HSDP VS. PGRAD

We compare two training methods for CA-TD: HSDP and PGRAD. Both use TT format with rank $R = 2$ and are tested on three Ackley grids. For each method, we also include unconstrained counterparts trained without constraint-awareness.

Table 2: Comparison of HSDP and PGRAD for constraint-aware surrogate training on discrete Ackley benchmarks. Each row reports the mean and standard deviation of the best objective value, the round at which it first appeared, and the average runtime per optimization round (in seconds).

| Task | Model | Constrained (-c) | | | Unconstrained (-u) | | |
|------|-------|------------|------------|-------------|------------|------------|-------------|
| | | Best Value | Best Round | Runtime (s) | Best Value | Best Round | Runtime (s) |
| Ackley | HSDP | $0.00 \pm 0.00$ | $3.40 \pm 0.92$ | $11.26 \pm 0.44$ | $1.32 \pm 1.32$ | $3.00 \pm 1.79$ | $11.49 \pm 0.32$ |
| $3 \times 3$ | PGRAD | $0.00 \pm 0.00$ | $3.70 \pm 1.55$ | $1.76 \pm 0.11$ | $0.00 \pm 0.00$ | $6.70 \pm 2.87$ | $2.03 \pm 0.49$ |
| Ackley | HSDP | $0.00 \pm 0.00$ | $6.90 \pm 2.02$ | $279.44 \pm 5.58$ | $0.00 \pm 0.00$ | $12.10 \pm 3.73$ | $229.56 \pm 0.78$ |
| $5 \times 5$ | PGRAD | $0.00 \pm 0.00$ | $5.60 \pm 2.97$ | $0.56 \pm 0.05$ | $0.00 \pm 0.00$ | $9.50 \pm 5.94$ | $1.44 \pm 0.85$ |
| Ackley | HSDP | $5.60 \pm 1.75$ | $1.30 \pm 0.46$ | $2045.80 \pm 65.16$ | $5.95 \pm 2.10$ | $1.00 \pm 0.00$ | $1932.85 \pm 33.06$ |
| $7 \times 7$ | PGRAD | $0.00 \pm 0.00$ | $12.60 \pm 4.94$ | $0.67 \pm 0.25$ | $0.00 \pm 0.00$ | $25.40 \pm 8.32$ | $1.10 \pm 0.45$ |

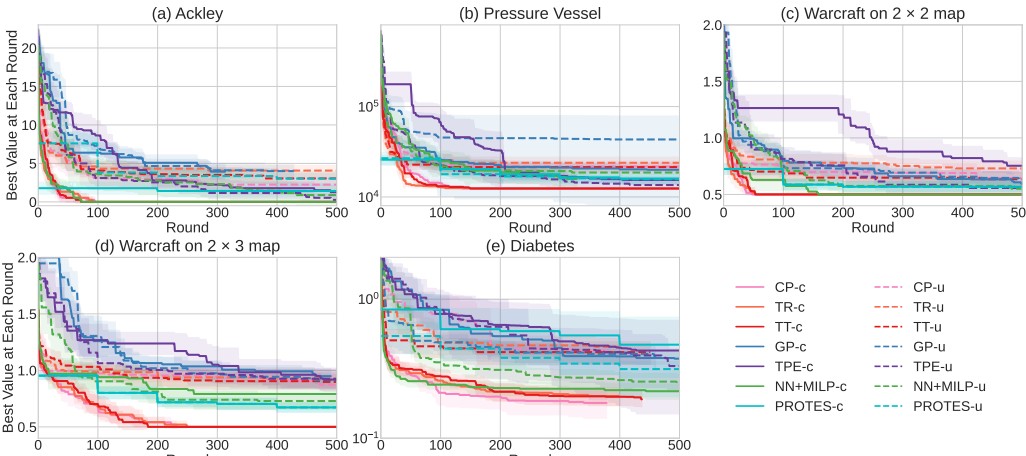

Figure 3: Optimization progress of our proposed tensor decomposition (TD) models and several baselines (GP, TPE, and NN+MILP), with and without constraint-awareness, across five benchmark tasks. For each TD model, the rank achieving the best performance is used (see Table 4 and Table 5 in the Appendix C for detailed results). Lower and earlier curves indicate better sample efficiency. Solid lines denote constrained models (-c). The enlarged view showing the separated parts -c and -u is shown in the Appendix I.

Figure 2 shows the optimization progress. On small grids, both constrained methods rapidly reach near-optimal values, with HSDP converging slightly earlier. On the $7 \times 7$ grid, HSDP becomes impractical due to computational cost, while PGRAD continues to improve efficiently. Table 2 summarizes best values, convergence rounds, and runtimes. PGRAD achieves strong performance across all cases with runtimes under one second per round. In contrast, HSDP is timed out (4000 seconds per round) in the case $7 \times 7$. Across all cases, constraint versions ("-c") outperform unconstrained ones ("-u") in both speed and final objective value.

## 5.4 EXPERIMENT 2: COMPARISON USING BENCHMARKS

In these experiments, we compare CA-TD with other methods using benchmarks. Also, to examine how different TD formats the performance of CA-TD, we evaluate it under three formats: Tensor Train (TT), Canonical Polyadic (CP) (Kolda & Bader, 2009), and Tensor Ring (TR) (Zhao et al., 2016), each tested at ranks $R = 2, \dots, 6$ (Appendix A). Throughout, constrained models are denoted with the suffix "-c", and unconstrained ones with "-u" where necessary. For TT and TR, the same rank $R$ is uniformly applied across all modes.

Figure 3 shows optimization curves for the best configuration of each method. Our proposed CA-TD models, e.g. TT-c, consistently achieve faster convergence and better final values compared to the naive baselines (GP-u, TPE-u, GP-c, and TPE-c). Crucially, CA-TD also demonstrates highly competitive or superior performance against the previous methods (PROTES and NN+MILP). This advantage is particularly evident in the Pressure Vessel and Warcraft benchmarks. Note that NN+MILP

is significantly affected by hyperparameters such as the number of training epochs, and here we use the best hyperparameters (Appendix D.3).

The CA-TD method also requires the appropriate selection of tensor format and rank. Our ablation study (Appendix D) reveals distinct characteristics for each tensor format. CP can achieve excellent final objective values, particularly on the Diabetes benchmark, but its performance is highly sensitive to rank, which varies widely across problems (from 2 to 6). TR exhibits exceptional convergence speed on the Pressure Vessel benchmark, yet its performance on other tasks is less competitive. In contrast, TT demonstrates the most consistent and robust performance across tasks, achieving the fastest convergence on the Ackley and Warcraft $2 \times 3$ benchmarks. Most importantly, the optimal rank of TT is remarkably stable, typically within the 3–4 range and reasonably consistent even at higher ranks, making it a user-friendly default choice for constrained BBO problems where ease of use and reliable performance are desired.

## 5.5 DISCUSSION

The experiments validate the effectiveness of CA-TD for constrained black-box optimization. From Experiment 1, we confirm that incorporating feasibility into the training step of the surrogate model improves sample efficiency. While HSDP performs well on small problems, PGRAD offers a scalable alternative suitable for larger settings such as the $7 \times 7$ Ackley grid.

From Experiment 2, we observe that our constraint-aware surrogate modeling is a dominant factor in improving performance. Our CA-TD model consistently outperforms unconstrained optimization methods such as GP-u and TPE-u, and methods that simply include prior information, such as GP-c and TPE-c, and performs comparably to more advanced NN+MILP(-c) methods. Among these, NN+MILP (both -c and -u) and GP-u/TPE-u are methods in which the surrogate model is trained without considering constraints, and constraints are considered only in the acquisition function stage. Our results suggest this leads to less efficient exploration. By training the surrogate model to learn the boundaries of the feasible space, CA-TD can more accurately predict promising regions and improve sample efficiency. This suggests that embedding feasibility directly into the surrogate model may be more effective than handling feasibility separately during acquisition.

Notably, our results show that CA-TD with the TT format delivers strong performance regardless of the tasks and the random seed and requires minimal hyperparameter tuning. This property highlights a key trade-off for practitioners between peak performance and practical usability. We believe that for general-purpose applications, where extensive preliminary analysis is not feasible, the robustness of the TT format provides a compelling advantage.

A key limitation of CA-TD lies in its further scalability to high-dimensional search spaces. While our experiments show speedups on PGRAD, further improvements in memory scalability are necessary to apply it to a wider range of problems. Tensor-based BBO, including CA-TD, is limited by the memory demands of dense tensor representations. Although a simple mini-batching strategy offers a preliminary workaround (Appendix H.1), fully scaling CA-TD to larger and higher-dimensional problems by using sparse tensor representation remains an important avenue for future research.

## 6 CONCLUSION

We proposed CA-TD, a constraint-aware surrogate modeling approach for sequential black-box optimization on discrete domains, integrating feasibility information directly into tensor decomposition–based surrogate models. We formulated the learning problem as a POP and introduced a relaxed algorithm, PGRAD, which showed competitive performance and superior scalability to larger problems compared with conventional HSDP. Experiments on synthetic and real-world inspired tasks, including an engineering design problem, indicated that CA-TD improves sample efficiency by incorporating constraints into the surrogate rather than into acquisition optimization. Future work includes scaling to higher-dimensional discrete spaces, e.g., via sparse tensor representations. While the TT format already mitigates scalability issues, automatic rank selection would further enhance the applicability of tensor formats. Extending the method to continuous domains and to BBO with constrained mixtures will also broaden its applications.

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

## A  FORMULATIONS OF TENSOR DECOMPOSITIONS

In our experiments (Section 5.4), we evaluated three tensor decomposition formats (CP, TR, and TT) with a rank parameter ($R \in \{2, \ldots, 6\}$) which controls model complexity. This appendix describes our formulation of each tensor decomposition form and how the parameter $R$ appears for each decomposition form.

In this paper, multiplication is defined as the operation that contraction (sum over) performs across adjacent indices. Formally, given two tensors $\mathbf{A}$ and $\mathbf{B}$, when contracting with respect to an index $i$, the tensor product is expressed as:

$$\mathbf{A}\mathbf{B} := \sum_i \mathbf{A}[\ldots, i]\mathbf{B}[i, \ldots],$$

where $\mathbf{A} \in \mathbb{R}^{\cdots \times |I|}$, $\mathbf{B} \in \mathbb{R}^{|I| \times \cdots}$, and $I$ denotes the range of the contraction index $i$ (for example, $i = 1, \ldots, |I|$).

**Canonical Polyadic (CP) Decomposition:**  The CP decomposition models a tensor as a sum of rank-one tensors. Its formulation is:

$$\hat{\mathcal{Y}}[x_1, \ldots, x_d] = \sum_{r=1}^{R} \mathbf{U}^{(1)}[x_1, r] \cdot \mathbf{U}^{(2)}[x_2, r] \cdots \mathbf{U}^{(d)}[x_d, r]$$

where $\mathbf{U}^{(k)} \in \mathbb{R}^{|X_k| \times R}$ is called the factor matrices and $\mathbf{U}^{(k)}[x_k, r]$ represents a scalar $(x_k, r)$-element in the $k$-th factor matrices. For the CP decomposition, the parameter $R$ used in our experiments directly corresponds to the rank of the decomposition, which is the number of rank-one tensors in the summation.

**Tensor Ring (TR) Decomposition:**  The TR decomposition represents a tensor as a circular product of third-order core tensors. Its formulation is:

$$\hat{\mathcal{Y}}[x_1, \ldots, x_d] = \text{Tr}(\mathbf{G}^{(1)}[x_1]\mathbf{G}^{(2)}[x_2] \cdots \mathbf{G}^{(d)}[x_d])$$

where $\mathbf{G}^{(k)}[x_k]$ is the $x_k$-th slice of the core tensor $\mathcal{G}^{(k)} \in \mathbb{R}^{r_{k-1} \times |X_k| \times r_k}$. Note that the trace operator $\text{Tr}(\cdot)$ is defined as $\text{Tr}(\mathbf{A}) := \sum_i \mathbf{A}[i, \ldots, i]$.

The complexity is characterized by a sequence of ranks $[r_1, r_2, \ldots, r_d]$ that form a cycle (i.e., $r_0 = r_d$). For our experiments, we applied a uniform rank setting, setting all TR-ranks to the common value $R$; i.e., $r_k = R$ for all $k \in \{1, \ldots, d\}$.

**Tensor-Train (TT) Decomposition:**  The TT decomposition, whose formulation is given in Eq. 1 in the main text, can be viewed as a special case of the TR decomposition. It effectively breaks the circular connection of the TR format by setting the boundary ranks to one ($r_0 = r_d = 1$). As with our TR experiments, we used a uniform rank setting for the internal ranks: $r_k = R$ for all $k \in \{1, \ldots, d - 1\}$.

## B  SPECIFICATIONS OF BENCHMARK PROBLEMS

This section provides details of the benchmark tasks used in our experiments. The differences in each search space are shown in Table 3. Details of each task are described below.

Table 3: Search spaces for each task

| task | search space size | #feasible points | ratio of feasible points |
|------|-------------------|------------------|--------------------------|
| Ackley 3×3 | $3^2$ | 5 | 0.56 |
| Ackley 5×5 | $5^2$ | 13 | 0.52 |
| Ackley 7×7 | $7^2$ | 29 | 0.59 |
| Ackley 65×65 | $65^2$ | 317 | 0.08 |
| Pressure Vessel | $10^4$ | 3916 | 0.39 |
| Warcraft 2×2 | $7^4$ | 300 | 0.12 |
| Warcraft 2×3 | $7^6$ | 5400 | 0.05 |
| Diabetes | $5^8$ | 10197 | 0.03 |

**Ackley**   The 2D Ackley function (Adorio & Diliman, 2005) on grid is defined as:

$$g(x_1, x_2) = -20 \exp\left(-0.2\sqrt{0.5(x_1^2 + x_2^2)}\right)$$
$$- \exp\left(0.5(\cos(2\pi x_1) + \cos(2\pi x_2))\right)$$
$$+ 20 + \exp(1).$$

The input space is discretized into a uniform integer grid, and feasibility is defined by a circular constraint $x_1^2 + x_2^2 \leq r^2$.

**Pressure Vessel**   The Pressure Vessel design problem is a classic engineering benchmark (Coello & Montes, 2002) with a mixed-variable search space. The goal is to minimize the total cost of a cylindrical pressure vessel. The problem has four variables, originally two continuous and two integer. For our experiments, we create a fully discrete search space by sampling 10 uniform levels from the domain of each variable. The objective function is given by:

$$g(x_1, x_2, x_3, x_4) = 0.6224x_1x_3x_4 + 1.7781x_2x_3^2$$
$$+ 3.1661x_1^2x_4 + 19.84x_1^2x_3,$$

subject to the following inequality constraints:

$$-x_1 + 0.0193x_3 \leq 0,$$
$$-x_2 + 0.00954x_3 \leq 0,$$
$$-\pi x_3^2 x_4 - \frac{4}{3}\pi x_3^3 + 1296000 \leq 0,$$
$$x_4 - 240 \leq 0.$$

**Warcraft**   This benchmark, adapted from a path prediction problem solved via supervised learning with combinatorial constraints presented in Ahmed et al. (2022), is treated as a black-box optimization task. The environment is a 2D $m \times n$ grid map, where each cell has a predefined traversal cost. An input $\mathbf{x} \in \mathcal{X}$, representing a candidate path, is encoded as a sequence of $m + n$ movement primitives. Each primitive is selected from seven movement types: vertical (up or down), horizontal (left or right), four L-shaped turns (e.g., up+right, regardless of order), and a null move (no displacement).

The objective function $g(\mathbf{x})$ evaluates each path by summing the traversal costs along the path and rewarding proximity to the bottom-right corner, with shorter Euclidean distance yielding better scores. The input is subject to three constraints: the path must start from the top-left cell, it must consist of exactly $m + n$ steps, and it must end at the bottom-right cell.

We evaluate two map sizes: $2 \times 2$ (path length 4) and $2 \times 3$ (path length 6), resulting in $7^4$ and $7^6$ candidate paths, respectively.

**Diabetes** This task simulates the goal of identifying actionable treatment plans for patients diagnosed with diabetes to demonstrate applicability on a real-world task where constraints are derived from domain knowledge. We use the Pima Indian Diabetes dataset (Smith et al., 1988), which contains 8 patient features and a binary diabetes label. Each continuous or integer-valued feature is discretized into 5 levels, resulting in a discrete search space $\mathcal{X} = \{0, 1, 2, 3, 4\}^8$.

A random forest classifier (Breiman, 2001) is trained on the entire dataset and used to predict the probability of diabetes for all candidates $\mathbf{x} \in \mathcal{X}$. Given a randomly chosen diabetic individual $\mathbf{x}_{\text{orig}}$, the goal is to find an alternative feature configuration $\mathbf{x} \in \mathcal{X}$ such that the predicted probability of diabetes is reduced. To encourage realistic plans, we penalize large deviations from the original configuration. Specifically, the objective function is defined as:

$$g(\mathbf{x}) = \text{RF}(\mathbf{x}) + \|\mathbf{x} - \mathbf{x}_{\text{orig}}\|_2,$$

where $\text{RF}(\mathbf{x}) \in [0, 1]$ is the predicted probability from the random forest classifier. Lower objective values correspond to medically plausible and effective treatment suggestions.

Feasibility constraints are imposed to exclude unrealistic feature combinations based on domain knowledge. For example, low insulin combined with high glucose is considered implausible for a non-diabetic patient and thus excluded from the feasible input.

## C EXPERIMENTAL DETAILS

All experiments were conducted on nodes running Ubuntu 22.04.5 LTS. Each experimental run was allocated 4 cores of an Intel Xeon Gold 6230R CPU and 8 GB of memory. A timeout of 3600 seconds (1 hour) was set for each run.

The software environment was built on Python 3.12.2. Key libraries include PyTorch 2.4.1 and NumPy 2.1.2. Our HSDP training strategy utilized ncpol2sdpa 1.12.2 and cvxpy 1.6.4, with SDPA 7.3.16 as the backend semidefinite programming solver. Our implementation of the NN+MILP baseline (Papalexopoulos et al., 2022) follows the methodology described in the original paper. The mixed-integer linear programming subproblems are solved using OR-Tools 9.14.6206.

**Details for the NN+MILP Baseline.** Following the original paper, our implementation of the NN+MILP(Papalexopoulos et al., 2022) uses a ReLU-based neural network consisting of one fully connected layer with 16 hidden dimensions as the surrogate model.

The acquisition problem, which seeks to maximize the surrogate's output, is formulated as an MILP. A key component of this formulation is the use of "no-good" cuts, which are constraints added to the MILP to exclude previously evaluated points from the search. This prevents the optimizer from repeatedly selecting the same points.

We tested two variants of this baseline:

- **Unconstrained (NN+MILP-u):** In this version, the MILP formulation only includes the search space boundaries and the "no-good" cuts as constraints. This variant does not use any specific knowledge about the problem's feasible region.
- **Constrained (NN+MILP-c):** This version extends the unconstrained setup by incorporating the explicit problem constraints directly into the MILP formulation. These are the same constraints used by our proposed CA-TD method, allowing for a direct and fair comparison of how constraint information is utilized. This includes, for example, the circular constraint for the Ackley problem and the physical constraints for the Pressure Vessel design.

## D ABLATION STUDY FOR EXPERIMENT 2

Table 4: Comprehensive comparison of CA-TD performance across tensor decomposition formats (CP, TR, TT) and tensor ranks on five benchmark tasks. For each configuration, we report the mean and standard deviation over ten runs for both the best feasible objective value found and the optimization round in which it first appeared. The best-performing rank for each model and task is identified based on the lowest average best value; in case of a tie, the configuration with the earlier average best round is selected. Selected best configurations are highlighted in **bold**.

| Objective | Constraint | Method | Rank | Best Value | Best Round |
|---|---|---|---|---|---|
| Ackley | Constrained | CP | 2 | $0.00 \pm 0.00$ | $69.40 \pm 14.00$ |
| | | | 3 | $\mathbf{0.00 \pm 0.00}$ | $\mathbf{37.70 \pm 26.38}$ |
| | | | 4 | $0.00 \pm 0.00$ | $50.20 \pm 33.32$ |
| | | | 5 | $0.00 \pm 0.00$ | $56.60 \pm 31.84$ |
| | | | 6 | $0.00 \pm 0.00$ | $63.50 \pm 24.45$ |
| | | TR | 2 | $0.00 \pm 0.00$ | $55.80 \pm 19.99$ |
| | | | 3 | $0.00 \pm 0.00$ | $61.60 \pm 36.76$ |
| | | | 4 | $\mathbf{0.00 \pm 0.00}$ | $\mathbf{48.60 \pm 29.46}$ |
| | | | 5 | $0.00 \pm 0.00$ | $57.70 \pm 34.97$ |
| | | | 6 | $0.00 \pm 0.00$ | $71.90 \pm 33.81$ |
| | | TT | 2 | $0.00 \pm 0.00$ | $58.00 \pm 20.76$ |
| | | | 3 | $\mathbf{0.00 \pm 0.00}$ | $\mathbf{36.30 \pm 19.66}$ |
| | | | 4 | $0.00 \pm 0.00$ | $47.50 \pm 24.36$ |
| | | | 5 | $0.00 \pm 0.00$ | $57.40 \pm 34.22$ |
| | | | 6 | $0.00 \pm 0.00$ | $63.70 \pm 41.13$ |
| | Unconstrained | CP | 2 | $5.00 \pm 2.59$ | $74.00 \pm 49.30$ |
| | | | 3 | $4.37 \pm 1.81$ | $179.50 \pm 158.71$ |
| | | | 4 | $5.11 \pm 1.51$ | $61.30 \pm 84.45$ |
| | | | 5 | $4.29 \pm 1.74$ | $76.80 \pm 76.16$ |
| | | | 6 | $\mathbf{2.24 \pm 1.97}$ | $\mathbf{105.10 \pm 102.10}$ |
| | | TR | 2 | $\mathbf{4.07 \pm 1.28}$ | $\mathbf{68.50 \pm 76.76}$ |
| | | | 3 | $5.72 \pm 1.54$ | $104.90 \pm 140.38$ |
| | | | 4 | $5.31 \pm 1.87$ | $47.80 \pm 49.07$ |
| | | | 5 | $5.01 \pm 2.05$ | $79.00 \pm 111.06$ |
| | | | 6 | $6.05 \pm 2.85$ | $27.30 \pm 22.45$ |
| | | TT | 2 | $4.74 \pm 1.96$ | $151.60 \pm 150.91$ |
| | | | 3 | $3.88 \pm 1.83$ | $135.70 \pm 139.12$ |
| | | | 4 | $3.70 \pm 1.77$ | $113.20 \pm 109.55$ |
| | | | 5 | $4.38 \pm 1.30$ | $111.30 \pm 89.04$ |
| | | | 6 | $\mathbf{3.05 \pm 1.30}$ | $\mathbf{138.40 \pm 131.81}$ |
| Diabetes | Constrained | CP | 2 | $0.26 \pm 0.08$ | $190.30 \pm 144.14$ |
| | | | 3 | $0.20 \pm 0.06$ | $245.40 \pm 105.69$ |
| | | | 4 | $0.22 \pm 0.05$ | $215.10 \pm 120.72$ |
| | | | 5 | $0.19 \pm 0.08$ | $229.20 \pm 121.43$ |
| | | | 6 | $\mathbf{0.18 \pm 0.08}$ | $\mathbf{183.40 \pm 92.49}$ |
| | | TR | 2 | $0.26 \pm 0.06$ | $193.50 \pm 134.80$ |
| | | | 3 | $0.25 \pm 0.04$ | $205.30 \pm 142.27$ |
| | | | 4 | $0.22 \pm 0.08$ | $276.70 \pm 134.96$ |
| | | | 5 | $0.20 \pm 0.06$ | $263.60 \pm 84.38$ |
| | | | 6 | $\mathbf{0.20 \pm 0.06}$ | $\mathbf{210.00 \pm 113.67}$ |
| | | TT | 2 | $0.27 \pm 0.05$ | $164.60 \pm 136.64$ |

(Table continues on next page)

Table 4: (Continued) Summary of Main Experiment Results

| Objective | Constraint | Method | Rank | Best Value | Best Round |
|---|---|---|---|---|---|
| | | | 3 | $0.25 \pm 0.06$ | $215.40 \pm 129.64$ |
| | | | 4 | $0.19 \pm 0.07$ | $289.50 \pm 77.59$ |
| | | | 5 | $0.21 \pm 0.06$ | $242.30 \pm 85.96$ |
| | | | 6 | $\mathbf{0.19 \pm 0.04}$ | $\mathbf{258.00 \pm 131.51}$ |
| | | CP | 2 | $0.56 \pm 0.19$ | $237.60 \pm 139.88$ |
| | | | 3 | $0.44 \pm 0.09$ | $105.70 \pm 120.23$ |
| | | | 4 | $0.49 \pm 0.15$ | $26.60 \pm 27.76$ |
| | | | 5 | $\mathbf{0.43 \pm 0.07}$ | $\mathbf{139.20 \pm 105.89}$ |
| | | | 6 | $0.44 \pm 0.11$ | $163.30 \pm 101.39$ |
| | Unconstrained | TR | 2 | $\mathbf{0.45 \pm 0.13}$ | $\mathbf{82.60 \pm 106.76}$ |
| | | | 3 | $0.49 \pm 0.20$ | $142.30 \pm 109.86$ |
| | | | 4 | $0.49 \pm 0.22$ | $21.80 \pm 30.35$ |
| | | | 5 | $0.59 \pm 0.20$ | $33.60 \pm 63.21$ |
| | | | 6 | $0.50 \pm 0.15$ | $62.80 \pm 75.78$ |
| | | TT | 2 | $0.59 \pm 0.17$ | $240.50 \pm 157.55$ |
| | | | 3 | $\mathbf{0.42 \pm 0.15}$ | $\mathbf{36.70 \pm 52.33}$ |
| | | | 4 | $0.43 \pm 0.13$ | $79.30 \pm 112.83$ |
| | | | 5 | $0.51 \pm 0.11$ | $24.30 \pm 46.99$ |
| | | | 6 | $0.43 \pm 0.15$ | $76.40 \pm 131.94$ |
| | | CP | 2 | $\mathbf{12408.34 \pm 0.00}$ | $\mathbf{97.60 \pm 22.90}$ |
| | | | 3 | $12408.34 \pm 0.00$ | $101.60 \pm 51.76$ |
| | | | 4 | $12408.34 \pm 0.00$ | $114.10 \pm 54.86$ |
| | | | 5 | $12408.34 \pm 0.00$ | $125.10 \pm 62.85$ |
| | | | 6 | $12408.34 \pm 0.00$ | $133.60 \pm 45.18$ |
| | Constrained | TR | 2 | $\mathbf{12408.34 \pm 0.00}$ | $\mathbf{65.60 \pm 39.50}$ |
| | | | 3 | $12408.34 \pm 0.00$ | $107.90 \pm 41.05$ |
| | | | 4 | $12408.34 \pm 0.00$ | $153.90 \pm 49.47$ |
| | | | 5 | $12408.34 \pm 0.00$ | $171.60 \pm 52.31$ |
| | | | 6 | $12408.34 \pm 0.00$ | $158.60 \pm 56.03$ |
| | | TT | 2 | $12408.34 \pm 0.00$ | $98.20 \pm 52.98$ |
| | | | 3 | $\mathbf{12408.34 \pm 0.00}$ | $\mathbf{91.80 \pm 41.26}$ |
| | | | 4 | $12408.34 \pm 0.00$ | $120.90 \pm 59.48$ |
| | | | 5 | $12408.34 \pm 0.00$ | $186.70 \pm 84.06$ |
| Pressure Vessel | | | 6 | $12408.34 \pm 0.00$ | $162.70 \pm 71.59$ |
| | | CP | 2 | $28747.63 \pm 6576.40$ | $135.10 \pm 144.70$ |
| | | | 3 | $26880.85 \pm 6015.81$ | $227.20 \pm 187.25$ |
| | | | 4 | $27551.16 \pm 9856.99$ | $166.20 \pm 183.89$ |
| | | | 5 | $24153.88 \pm 4083.86$ | $63.90 \pm 90.89$ |
| | | | 6 | $\mathbf{21718.03 \pm 6122.11}$ | $\mathbf{83.80 \pm 60.81}$ |
| | Unconstrained | TR | 2 | $28563.29 \pm 6355.42$ | $42.20 \pm 30.29$ |
| | | | 3 | $24871.78 \pm 7397.81$ | $65.80 \pm 68.65$ |
| | | | 4 | $32783.74 \pm 8703.58$ | $36.00 \pm 35.68$ |
| | | | 5 | $34158.80 \pm 6932.70$ | $19.20 \pm 12.46$ |
| | | | 6 | $\mathbf{23866.31 \pm 7088.02}$ | $\mathbf{28.70 \pm 32.95}$ |
| | | | 2 | $26976.59 \pm 4829.46$ | $36.70 \pm 53.05$ |
| | | | 3 | $26170.20 \pm 7600.21$ | $65.20 \pm 129.01$ |
| | | TT | | (Table continues on next page) | |

Table 4: (Continued) Summary of Main Experiment Results

| Objective | Constraint | Method | Rank | Best Value | Best Round |
|---|---|---|---|---|---|
| | | | 4 | $22564.40 \pm 5990.01$ | $67.70 \pm 90.85$ |
| | | | 5 | $22935.84 \pm 4399.48$ | $28.40 \pm 33.67$ |
| | | | 6 | $\mathbf{21395.48 \pm 4376.77}$ | $\mathbf{34.60 \pm 26.78}$ |
| Warcraft $2 \times 2$ | Constrained | CP | 2 | $0.50 \pm 0.00$ | $40.80 \pm 40.34$ |
| | | | 3 | $0.50 \pm 0.00$ | $31.00 \pm 22.28$ |
| | | | 4 | $0.50 \pm 0.00$ | $31.30 \pm 14.60$ |
| | | | 5 | $0.50 \pm 0.00$ | $37.90 \pm 15.57$ |
| | | | 6 | $\mathbf{0.50 \pm 0.00}$ | $\mathbf{25.60 \pm 15.50}$ |
| | | TR | 2 | $0.50 \pm 0.00$ | $29.60 \pm 18.17$ |
| | | | 3 | $0.50 \pm 0.00$ | $29.70 \pm 16.46$ |
| | | | 4 | $0.50 \pm 0.00$ | $33.00 \pm 12.57$ |
| | | | 5 | $\mathbf{0.50 \pm 0.00}$ | $\mathbf{26.10 \pm 18.61}$ |
| | | | 6 | $0.50 \pm 0.00$ | $41.80 \pm 16.80$ |
| | | TT | 2 | $0.50 \pm 0.00$ | $34.90 \pm 21.39$ |
| | | | 3 | $\mathbf{0.50 \pm 0.00}$ | $\mathbf{34.10 \pm 14.74}$ |
| | | | 4 | $0.50 \pm 0.00$ | $39.80 \pm 22.00$ |
| | | | 5 | $0.50 \pm 0.00$ | $41.60 \pm 16.08$ |
| | | | 6 | $0.50 \pm 0.00$ | $47.20 \pm 21.19$ |
| | Unconstrained | CP | 2 | $0.67 \pm 0.12$ | $192.30 \pm 157.25$ |
| | | | 3 | $0.74 \pm 0.08$ | $108.90 \pm 99.23$ |
| | | | 4 | $0.68 \pm 0.18$ | $212.00 \pm 189.91$ |
| | | | 5 | $\mathbf{0.65 \pm 0.14}$ | $\mathbf{96.30 \pm 132.72}$ |
| | | | 6 | $0.68 \pm 0.12$ | $181.20 \pm 197.93$ |
| | | TR | 2 | $\mathbf{0.73 \pm 0.12}$ | $\mathbf{171.40 \pm 138.09}$ |
| | | | 3 | $0.74 \pm 0.11$ | $188.20 \pm 181.50$ |
| | | | 4 | $0.74 \pm 0.10$ | $163.90 \pm 133.61$ |
| | | | 5 | $0.76 \pm 0.12$ | $123.70 \pm 146.50$ |
| | | | 6 | $0.76 \pm 0.07$ | $195.30 \pm 152.81$ |
| | | TT | 2 | $0.65 \pm 0.11$ | $192.30 \pm 147.74$ |
| | | | 3 | $0.75 \pm 0.13$ | $213.00 \pm 172.45$ |
| | | | 4 | $0.71 \pm 0.09$ | $181.40 \pm 181.45$ |
| | | | 5 | $0.67 \pm 0.13$ | $195.80 \pm 170.67$ |
| | | | 6 | $\mathbf{0.63 \pm 0.12}$ | $\mathbf{115.60 \pm 147.26}$ |
| | Constrained | CP | 2 | $0.56 \pm 0.09$ | $108.10 \pm 53.07$ |
| | | | 3 | $0.50 \pm 0.00$ | $145.20 \pm 70.85$ |
| | | | 4 | $\mathbf{0.50 \pm 0.00}$ | $\mathbf{123.00 \pm 68.13}$ |
| | | | 5 | $0.50 \pm 0.00$ | $125.70 \pm 67.57$ |
| | | | 6 | $0.50 \pm 0.00$ | $126.90 \pm 66.16$ |
| | | TR | 2 | $0.56 \pm 0.09$ | $335.20 \pm 118.54$ |
| | | | 3 | $\mathbf{0.50 \pm 0.00}$ | $\mathbf{139.10 \pm 60.37}$ |
| | | | 4 | $0.51 \pm 0.03$ | $255.50 \pm 147.65$ |
| | | | 5 | $0.50 \pm 0.00$ | $144.50 \pm 90.58$ |
| | | | 6 | $0.50 \pm 0.00$ | $175.20 \pm 95.16$ |
| | | TT | 2 | $0.58 \pm 0.10$ | $257.90 \pm 117.20$ |
| | | | 3 | $0.50 \pm 0.00$ | $116.90 \pm 30.93$ |
| | | | 4 | $\mathbf{0.50 \pm 0.00}$ | $\mathbf{115.70 \pm 34.39}$ |

(Table continues on next page)

Warcraft $2 \times 3$

Table 4: (Continued) Summary of Main Experiment Results

| Objective | Constraint | Method | Rank | Best Value | Best Round |
|---|---|---|---|---|---|
| | | | 5 | $0.50 \pm 0.00$ | $144.70 \pm 40.49$ |
| | | | 6 | $0.50 \pm 0.00$ | $152.80 \pm 42.79$ |
| | | | 2 | $0.97 \pm 0.08$ | $221.70 \pm 175.95$ |
| | | | 3 | $\mathbf{0.87 \pm 0.10}$ | $\mathbf{234.50 \pm 192.92}$ |
| | | CP | 4 | $0.90 \pm 0.13$ | $96.90 \pm 80.52$ |
| | | | 5 | $0.88 \pm 0.14$ | $148.50 \pm 132.13$ |
| | | | 6 | $0.88 \pm 0.12$ | $156.10 \pm 116.36$ |
| | | | 2 | $\mathbf{0.93 \pm 0.15}$ | $\mathbf{130.20 \pm 159.53}$ |
| | | | 3 | $1.01 \pm 0.06$ | $92.40 \pm 118.96$ |
| | Unconstrained | TR | 4 | $1.04 \pm 0.06$ | $150.30 \pm 161.71$ |
| | | | 5 | $0.94 \pm 0.11$ | $120.90 \pm 127.46$ |
| | | | 6 | $0.93 \pm 0.11$ | $91.50 \pm 142.73$ |
| | | | 2 | $\mathbf{0.90 \pm 0.13}$ | $\mathbf{184.60 \pm 128.92}$ |
| | | | 3 | $0.95 \pm 0.10$ | $204.20 \pm 172.71$ |
| | | TT | 4 | $0.98 \pm 0.07$ | $167.80 \pm 148.80$ |
| | | | 5 | $1.01 \pm 0.06$ | $148.90 \pm 154.59$ |
| | | | 6 | $0.96 \pm 0.12$ | $56.90 \pm 76.03$ |

Table 5: Performance summary of the baseline methods (GP, TPE, NN+MILP, and PROTES) corresponding to the optimization progress in Experiment 2. For each configuration, we report the mean and standard deviation over ten runs for both the best feasible objective value found and the optimization round in which it first appeared. The best-performing method for each task and constraint is identified based on the lowest average best value.

| Objective | Constraint | Method | Best Value | Best Round |
|---|---|---|---|---|
| Ackley | Constrained | GP | $3.72 \pm 1.52$ | $249.90 \pm 105.64$ |
| | | TPE | $1.25 \pm 1.57$ | $295.50 \pm 109.26$ |
| | | NN+MILP | $0.00 \pm 0.00$ | $41.30 \pm 24.93$ |
| | | PROTES | $1.42 \pm 1.44$ | $140.00 \pm 80.00$ |
| | Unconstrained | GP | $4.00 \pm 1.55$ | $209.40 \pm 90.30$ |
| | | TPE | $0.26 \pm 0.79$ | $254.10 \pm 162.61$ |
| | | NN+MILP | $0.88 \pm 2.65$ | $222.30 \pm 167.06$ |
| | | PROTES | $3.03 \pm 1.70$ | $280.00 \pm 124.90$ |
| Diabetes | Constrained | GP | $0.37 \pm 0.07$ | $262.20 \pm 149.81$ |
| | | TPE | $0.43 \pm 0.53$ | $312.30 \pm 126.52$ |
| | | NN+MILP | $0.22 \pm 0.05$ | $176.60 \pm 134.49$ |
| | | PROTES | $0.47 \pm 0.52$ | $410.00 \pm 122.07$ |
| | Unconstrained | GP | $0.39 \pm 0.13$ | $208.80 \pm 158.22$ |
| | | TPE | $0.33 \pm 0.20$ | $388.80 \pm 92.61$ |
| | | NN+MILP | $0.25 \pm 0.07$ | $334.30 \pm 122.75$ |
| | | PROTES | $0.31 \pm 0.08$ | $380.00 \pm 124.90$ |
| Pressure Vessel | Constrained | GP | $19982.64 \pm 3972.10$ | $250.30 \pm 155.75$ |
| | | TPE | $15375.32 \pm 2757.68$ | $294.10 \pm 66.56$ |
| | | NN+MILP | $14394.76 \pm 3905.88$ | $239.20 \pm 137.69$ |
| | | PROTES | $16006.94 \pm 3748.15$ | $350.00 \pm 111.80$ |

(Table continues on next page)

Table 5: (Continued) Summary of Baseline Method Results

| Objective | Constraint | Method | Best Value | Best Round |
|---|---|---|---|---|
| | Unconstrained | GP | $43256.59 \pm 67170.11$ | $169.70 \pm 147.72$ |
| | | TPE | $13550.90 \pm 1521.29$ | $238.20 \pm 104.85$ |
| | | NN+MILP | $18632.64 \pm 2319.82$ | $161.10 \pm 88.91$ |
| | | PROTES | $15704.22 \pm 3402.14$ | $340.00 \pm 101.98$ |
| Warcraft $2 \times 2$ | Constrained | GP | $0.59 \pm 0.10$ | $307.00 \pm 153.05$ |
| | | TPE | $0.75 \pm 0.13$ | $288.30 \pm 129.27$ |
| | | NN+MILP | $0.50 \pm 0.00$ | $66.10 \pm 59.91$ |
| | | PROTES | $0.57 \pm 0.08$ | $190.00 \pm 53.85$ |
| | Unconstrained | GP | $0.64 \pm 0.10$ | $271.50 \pm 136.55$ |
| | | TPE | $0.56 \pm 0.11$ | $194.10 \pm 105.68$ |
| | | NN+MILP | $0.55 \pm 0.10$ | $150.60 \pm 119.96$ |
| | | PROTES | $0.57 \pm 0.08$ | $190.00 \pm 53.85$ |
| Warcraft $2 \times 3$ | Constrained | GP | $0.95 \pm 0.13$ | $225.60 \pm 154.90$ |
| | | TPE | $0.92 \pm 0.19$ | $280.10 \pm 159.76$ |
| | | NN+MILP | $0.79 \pm 0.26$ | $85.00 \pm 98.29$ |
| | | PROTES | $0.67 \pm 0.08$ | $360.00 \pm 128.06$ |
| | Unconstrained | GP | $0.91 \pm 0.13$ | $231.30 \pm 98.45$ |
| | | TPE | $0.93 \pm 0.16$ | $169.50 \pm 130.75$ |
| | | NN+MILP | $0.73 \pm 0.12$ | $151.40 \pm 101.18$ |
| | | PROTES | $0.67 \pm 0.08$ | $360.00 \pm 128.06$ |

## D.1 Effect of the Penalty Coefficient $\lambda$

We conduct an ablation study to assess the sensitivity of our PGRAD strategy to the penalty coefficient $\lambda$, which was fixed at 1.0 in our main experiments. We test the CP-c, TR-c, and TT-c surrogates on all five benchmarks, varying $\lambda$ across the range $\{10, 5, 1, 0.5, 0.1, 0.01, 0.001, 0.0001\}$ for each benchmark.

The results, presented in Figures 4–6, demonstrate that the performance of these surrogates is remarkably robust to this hyperparameter across all benchmarks and tensor ranks. Within each subfigure, the plots compare the optimization progress over rounds for different values of $\lambda$, where lower values indicate better performance. For all tested values of $\lambda$, the optimization progress curves are nearly identical, showing similar convergence behavior. Furthermore, most runs completed their full evaluation budget. A minor exception was observed on the Warcraft $2\times3$ map benchmark, where the highest penalty coefficients ($\lambda = 10$ and $\lambda = 5$) resulted in slightly worse performance for all three methods (CP-c, TR-c, TT-c) In our main experiments, the constrained GP and TPE baselines (GP-c and TPE-c) were trained with 200 offline-sampled infeasible inputs. Here, we conduct an ablation study to analyze how the number of these infeasible points affects their performance. We test on all five benchmarks, varying the number of infeasible points in $\{0, 50, 100, 200, 300, 500, 1000, 2000\}$.

The results are presented in Figure 7. We observe a consistent and counter-intuitive trend across all benchmarks: for both GP-c and TPE-c, increasing the number of pre-trained infeasible points generally leads to a degradation in optimization performance. This effect is particularly severe for the GP baseline. As the number of informed points increases, the GP model's performance consistently worsens, and the computational overhead leads to premature termination of the optimization process, evidenced by truncated convergence curves. This is likely because the GP model, which scales cubically with the number of data points, becomes prohibitively expensive to train and use for acquisition function optimization.

The TPE baseline, while also showing some performance degradation with more infeasible points, proves to be more computationally robust and completes its evaluation budget in most cases. These

findings suggest that naively informing baseline models about the infeasible space by simply expanding the training dataset is not an effective strategy and can be detrimental, especially for computationally intensive models. This highlights the need for more sophisticated constraint-handling methods.

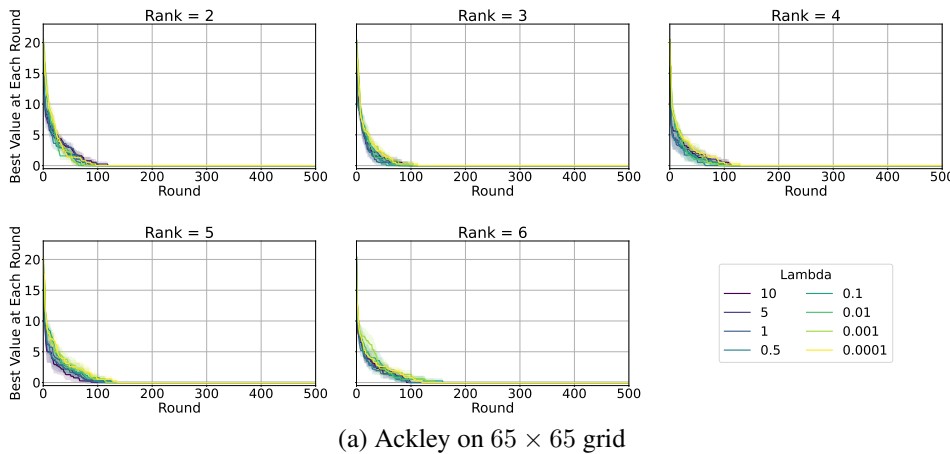

(a) Ackley on $65 \times 65$ grid

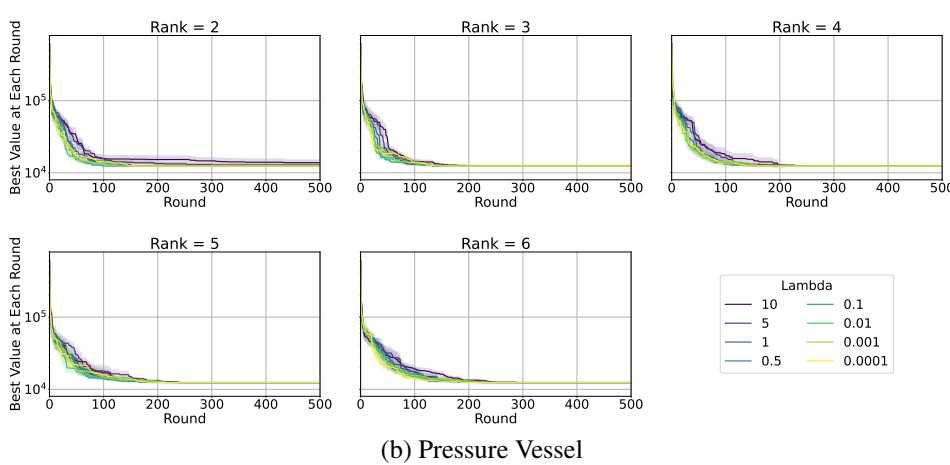

(b) Pressure Vessel

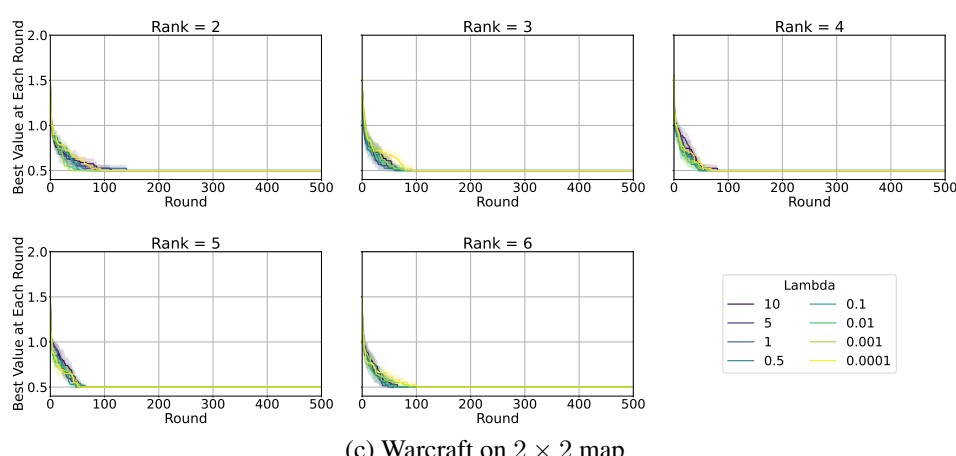

(c) Warcraft on $2 \times 2$ map

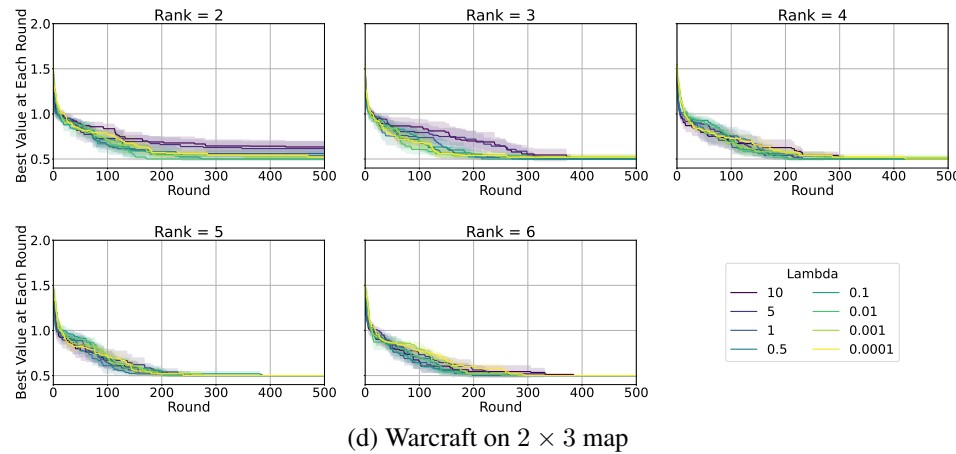

(d) Warcraft on $2 \times 3$ map

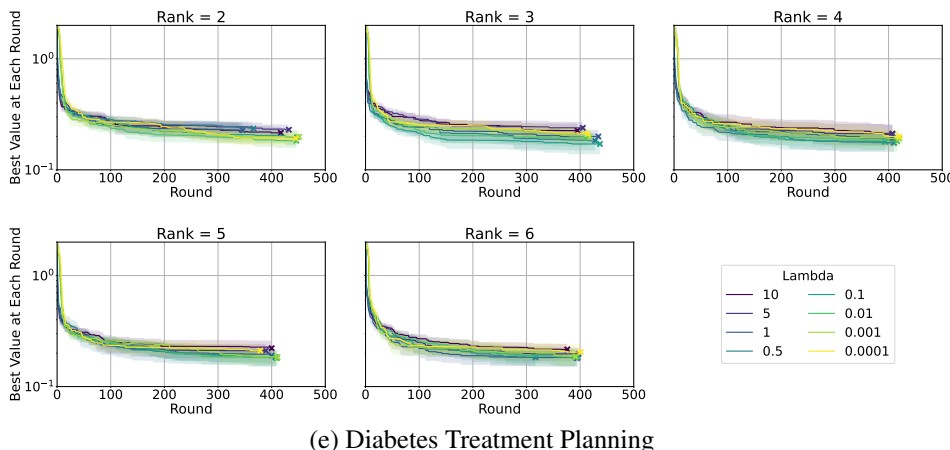

(e) Diabetes Treatment Planning

Figure 4: Ablation study for CP-c across all five benchmarks. Each panel shows the optimization progress for a different tensor rank, comparing various settings of the penalty coefficient $\lambda$.

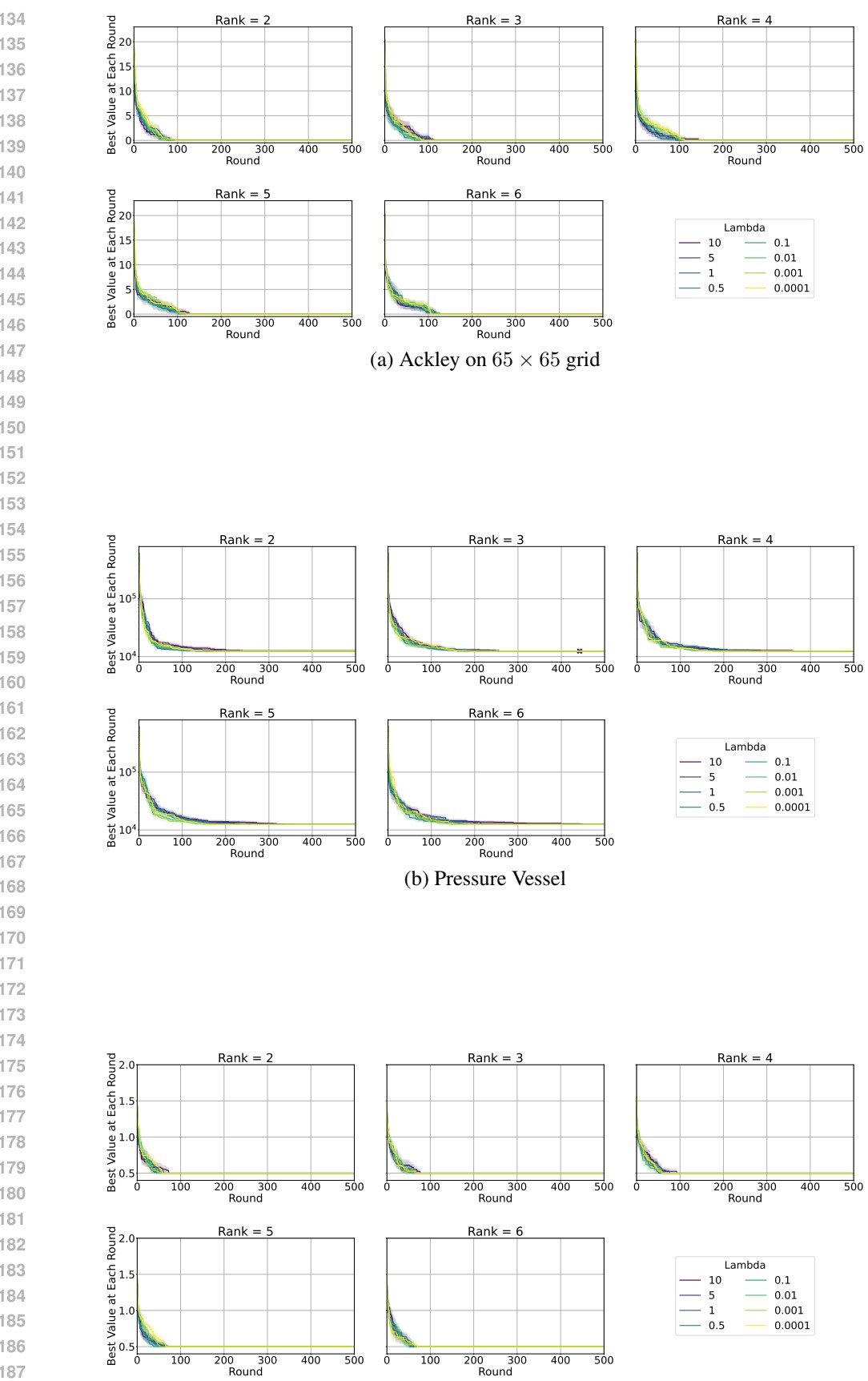

(a) Ackley on $65 \times 65$ grid

(b) Pressure Vessel

(c) Warcraft on $2 \times 2$ map

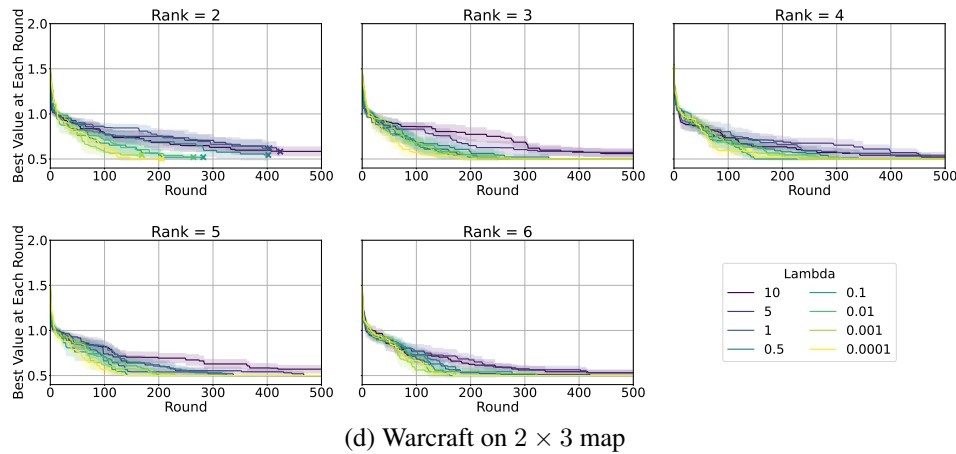

(d) Warcraft on $2 \times 3$ map

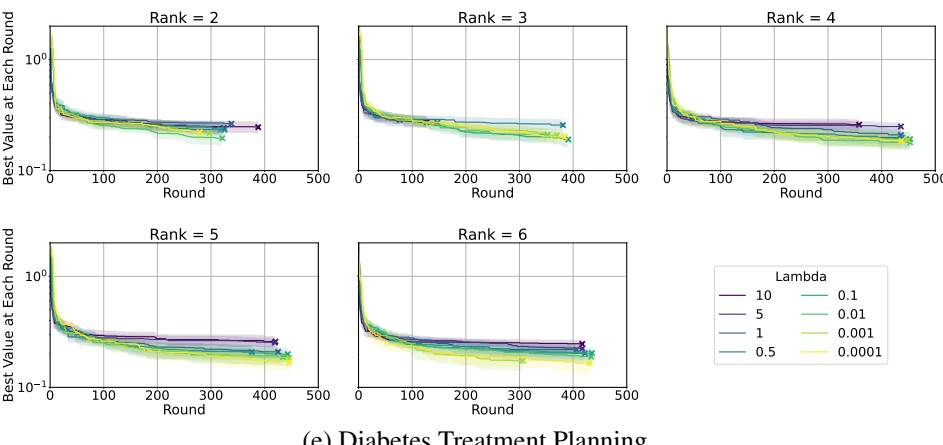

(e) Diabetes Treatment Planning

Figure 5: Ablation study for the TR-c method across all five benchmarks. Each panel shows the optimization progress for a different tensor rank, comparing various settings of the penalty coefficient $\lambda$.

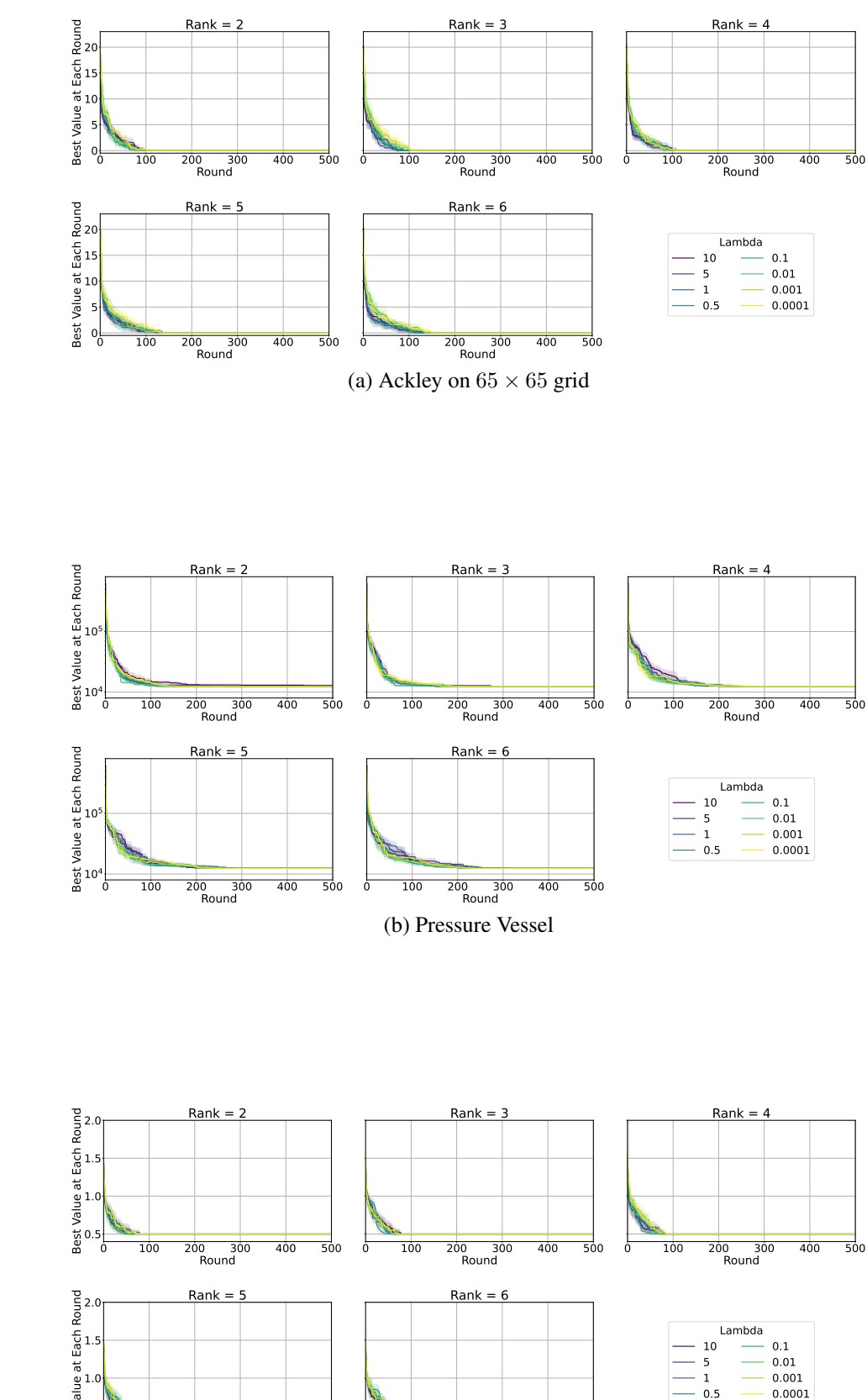

(a) Ackley on $65 \times 65$ grid

(b) Pressure Vessel

(c) Warcraft on $2 \times 2$ map

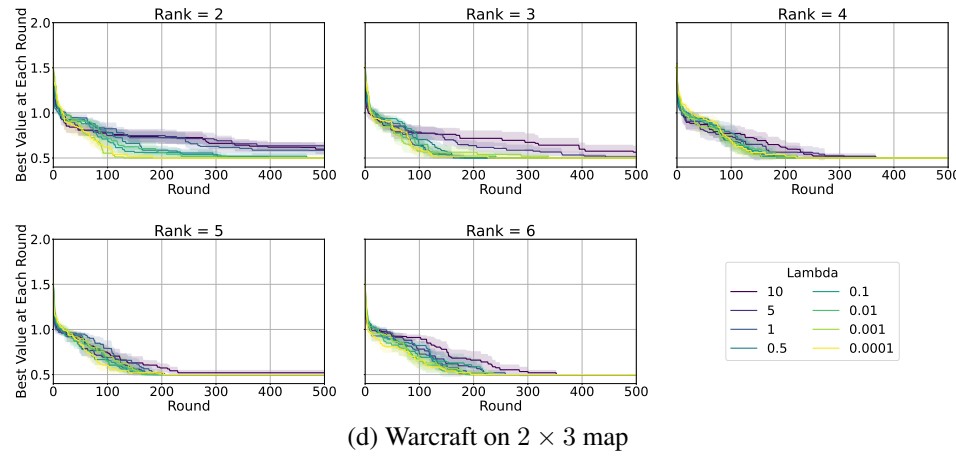

(d) Warcraft on $2 \times 3$ map

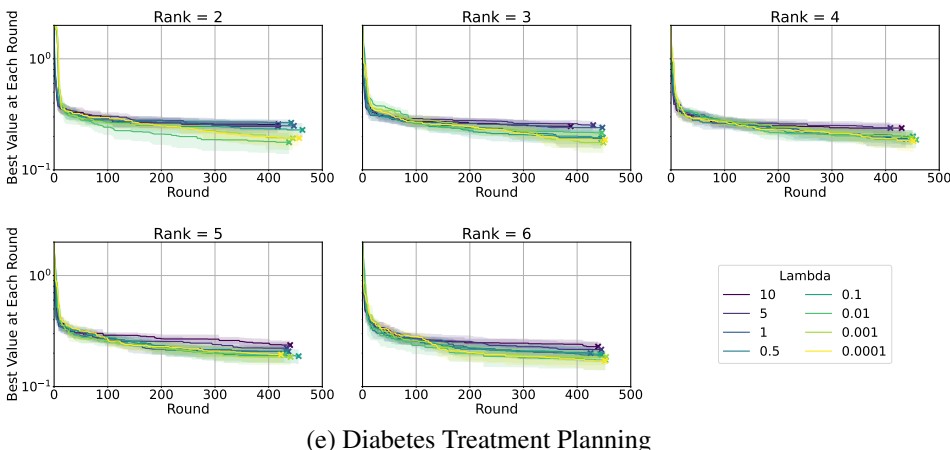

(e) Diabetes Treatment Planning

Figure 6: Ablation study for TT-c across all five benchmarks. Each panel shows the optimization progress for a different tensor rank, comparing various settings of the penalty coefficient $\lambda$.

## D.2 EFFECT OF OFFLINE TRAINING ON GP AND TPE BASELINES

In our main experiments, the constrained GP and TPE baselines (GP-c and TPE-c) were trained with 200 offline-sampled infeasible inputs. Here, we conduct an ablation study to analyze how the number of these infeasible points affects their performance. We test on all five benchmarks, varying the number of infeasible points in $\{0, 50, 100, 200, 300, 500, 1000, 2000\}$.

The results are presented in Figure 7. We observe a consistent and counter-intuitive trend across all benchmarks: for both GP-c and TPE-c, increasing the number of pre-trained infeasible points generally leads to a degradation in optimization performance. This effect is particularly severe for the GP baseline. As the number of informed points increases, the GP model's performance consistently worsens, and the computational overhead leads to premature termination of the optimization process, evidenced by truncated convergence curves. This is likely because the GP model, which scales cubically with the number of data points, becomes prohibitively expensive to train and use for acquisition function optimization.

The TPE baseline, while also showing some performance degradation with more infeasible points, proves to be more computationally robust and completes its evaluation budget in most cases. These findings suggest that naively informing baseline models about the infeasible space by simply expanding the training dataset is not an effective strategy and can be detrimental, especially for computationally intensive models. This highlights the need for more sophisticated constraint-handling methods.

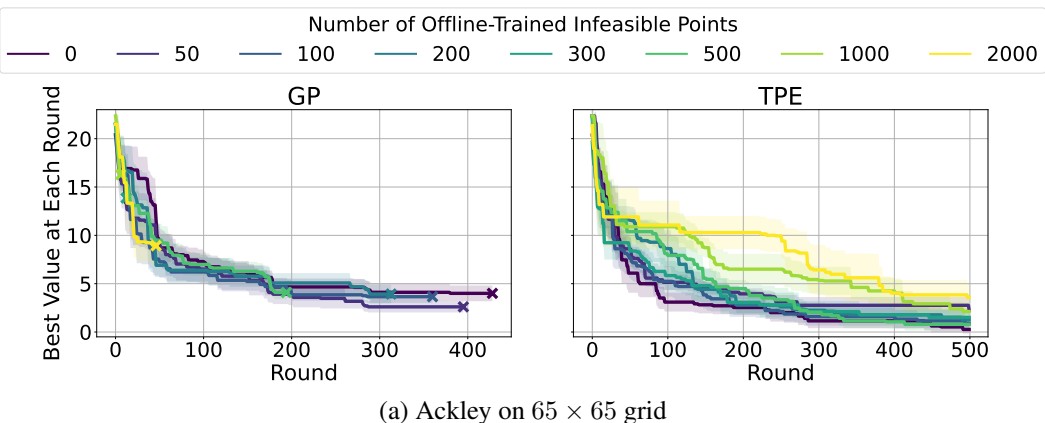

(a) Ackley on $65 \times 65$ grid

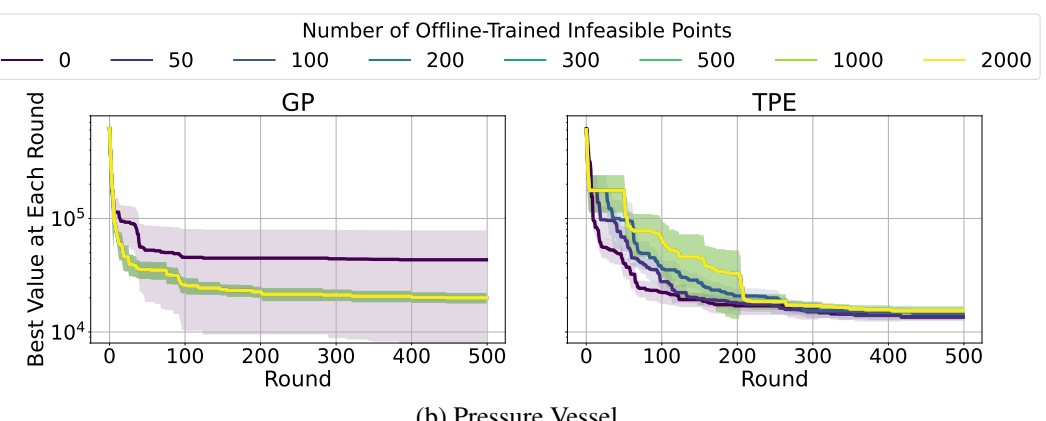

(b) Pressure Vessel

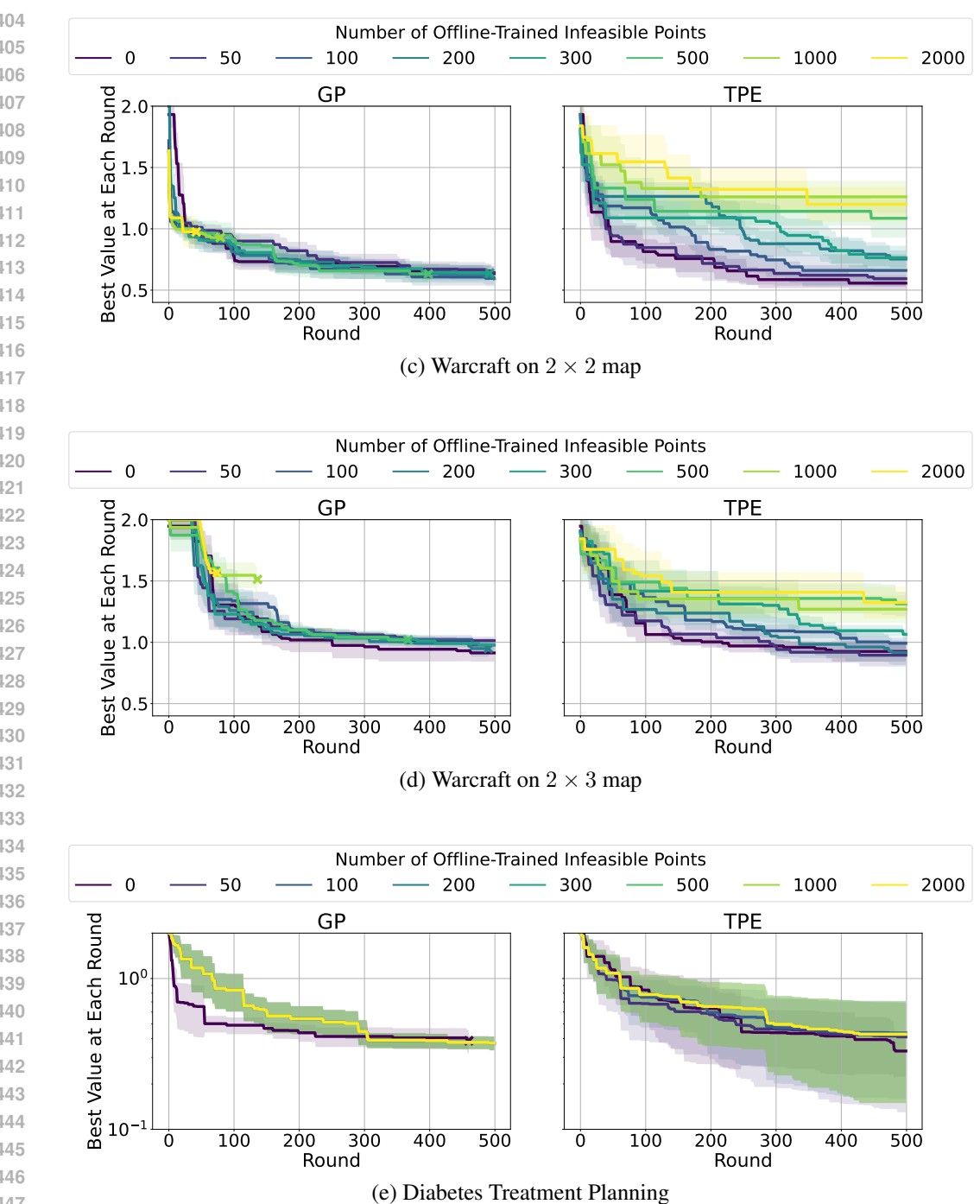

(c) Warcraft on $2 \times 2$ map

(d) Warcraft on $2 \times 3$ map

(e) Diabetes Treatment Planning

Figure 7: Ablation study on the number of pre-trained infeasible points for GP-c and TPE-c baselines across all five benchmarks. Each row corresponds to a benchmark, with the left and right panels showing the performance of GP-c and TPE-c, respectively. The results demonstrate that increasing the number of informed points generally degrades performance for both methods.

### D.3 EFFECT OF NUMBER OF EPOCHS AND INITIAL POINTS FOR NN+MILP

To determine a hyperparameter configuration for the NN+MILP baseline, we conduct an ablation study on its two key parameters: the number of training epochs for the neural network surrogate and the number of initial random points. We tested the number of epochs over the set $\{100, 300, 1000, 5000, 10000, 25000\}$ and evaluated using either 1 or 50 initial points.

The results for all five benchmarks are presented in Figure 8. From these figures, we can draw several conclusions to guide our selection.

First, regarding the number of initial points, a clear distinction emerges depending on the presence of constraints. For the constrained setting (-c, solid lines), using a single initial point (blue solid lines) demonstrates superior or competitive performance compared to 50 initial points (orange solid lines) across most tasks. This trend is especially noticeable in the (a) Ackley and (c) Warcraft $2 \times 3$ benchmarks. Conversely, in the unconstrained setting (-u, dashed lines), using 50 initial points (orange dashed lines) leads to significantly faster convergence and better final performance than a single point. This suggests that the optimal number of initial points is contingent on whether constraints are applied.

Second, concerning the number of training epochs, performance consistently improves up to 1000 epochs. Beyond this point, for instance, at 5000 or 10000 epochs, we observe diminishing returns; the significant increase in computational cost does not yield a correspondingly large improvement in optimization performance. This suggests that, for the scale of problems considered in our study, approximately 1000 epochs provide a sufficient training budget for the surrogate model.

Based on this analysis, we conclude that the optimal configuration depends on the constraint setting: 1 initial point with 1000 training epochs for constrained problems (-c), and 50 initial points with 1000 training epochs for unconstrained problems (-u). Each of these configurations offers the best trade-off between sample efficiency, final performance, and computational cost for its respective context. Therefore, we adopt these respective settings for the NN+MILP baseline in all main experiments presented in Section 5.

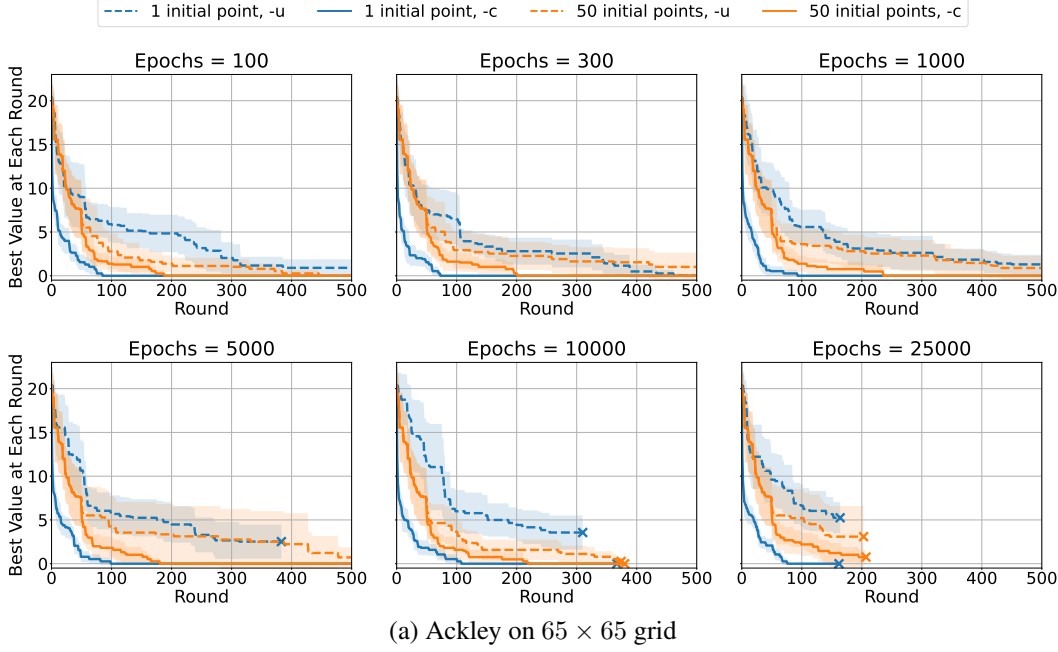

(a) Ackley on $65 \times 65$ grid

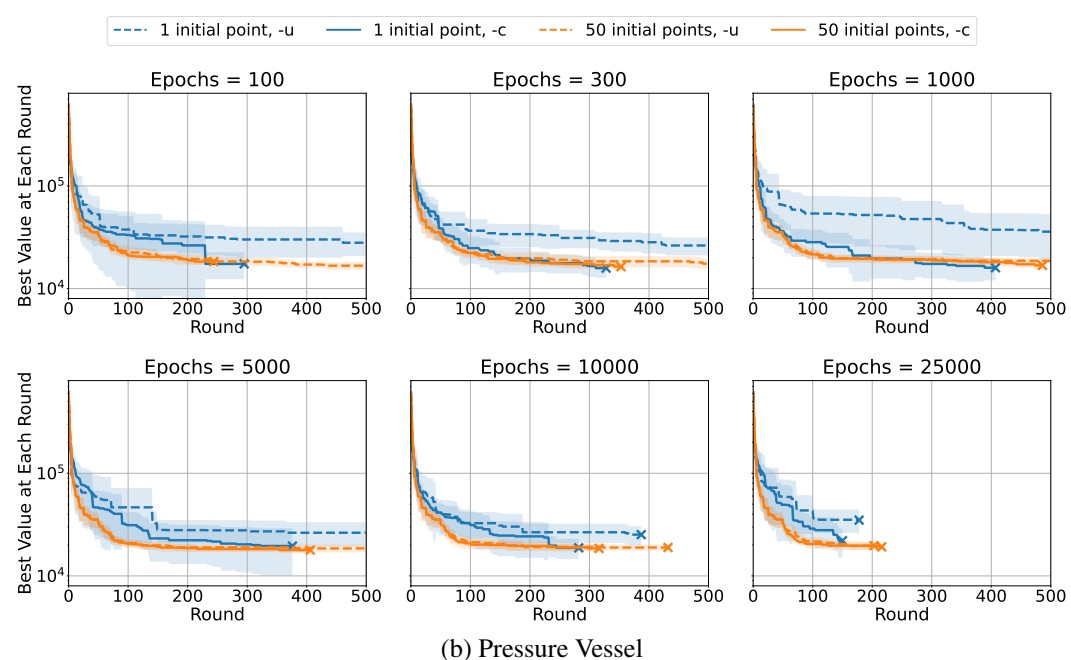

(b) Pressure Vessel

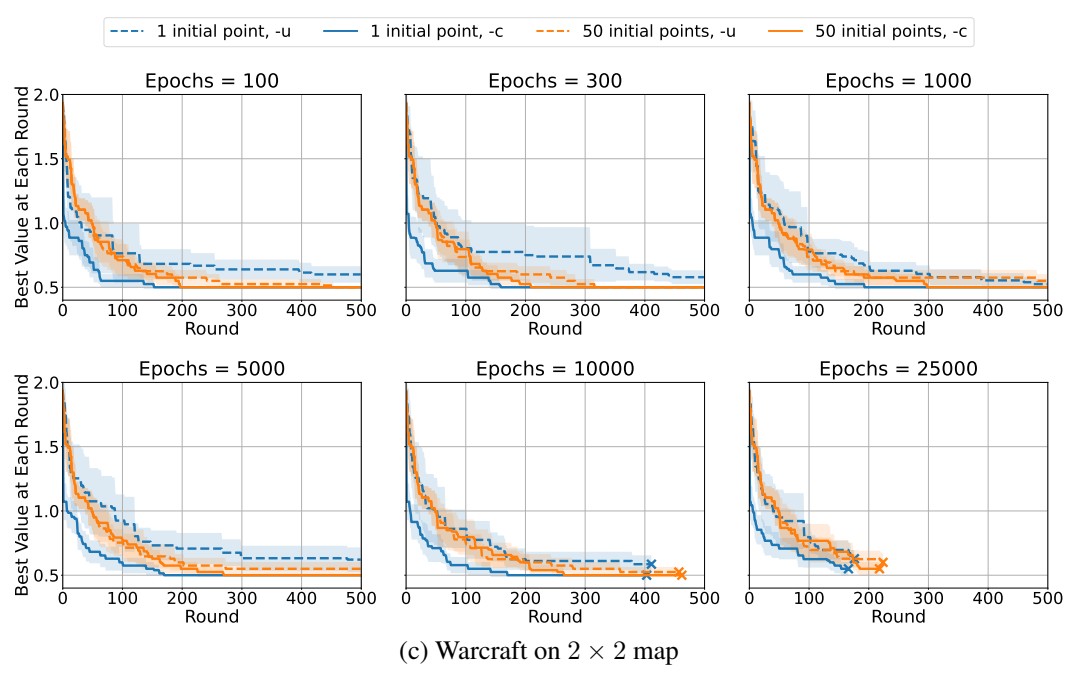

(c) Warcraft on $2 \times 2$ map

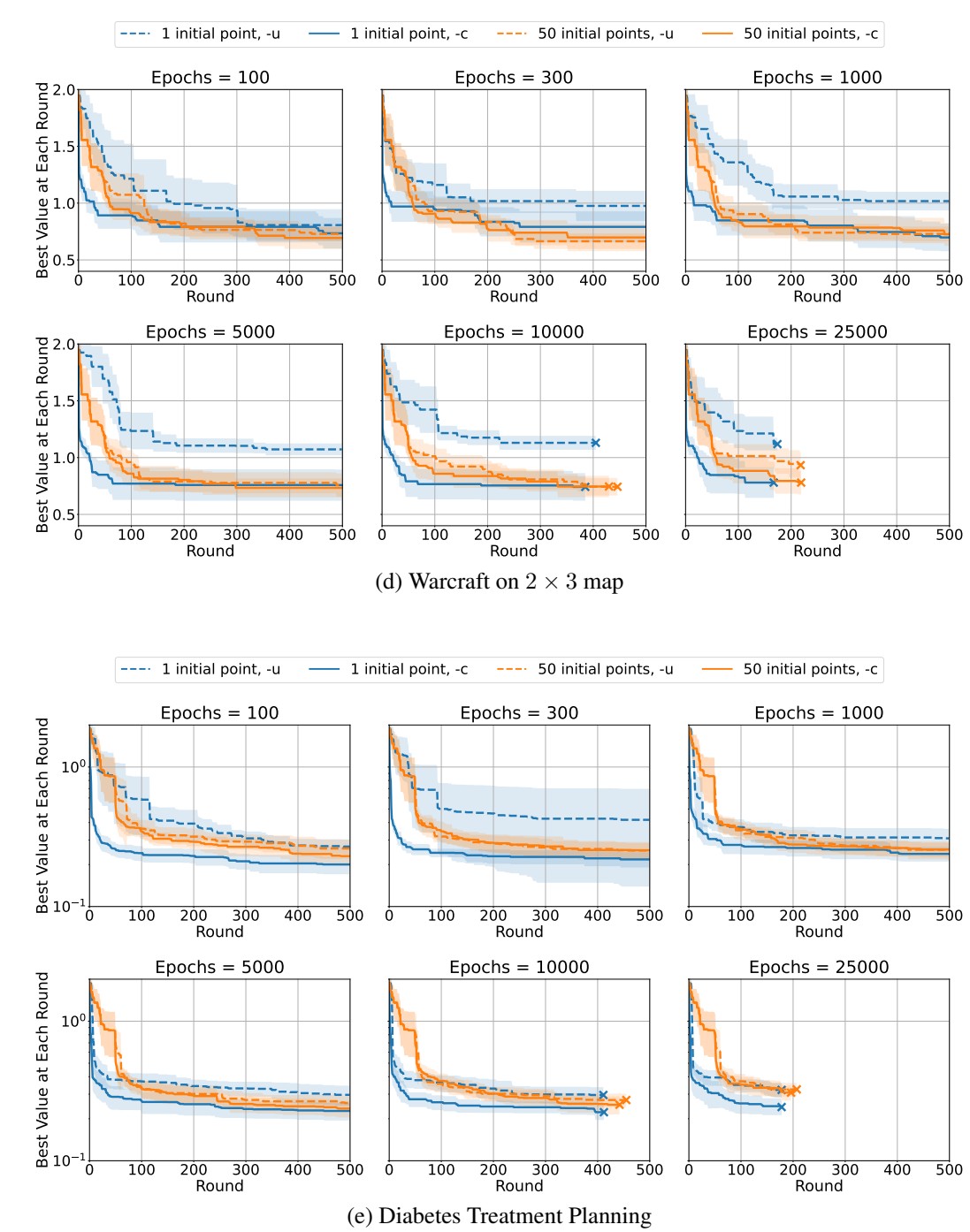

(d) Warcraft on $2 \times 3$ map

(e) Diabetes Treatment Planning

Figure 8: Ablation study for the NN+MILP baseline across all five benchmarks: (a) Ackley, (b) Pressure Vessel, (c) Warcraft $2 \times 2$, (d) Warcraft $2 \times 3$, and (e) Diabetes. Each panel within a benchmark's plot shows the optimization progress for a different number of training epochs. Blue and orange lines correspond to using 1 and 50 initial random points, respectively.

## D.4 Effect of Hyperparameters for PROTES

In each optimization round, PROTES evaluates a batch of $B$ points on the objective function. From this batch, the top $K$ samples are selected to train the TT surrogate model. For this training, the number of iterations for the gradient-based method is fixed at 100 for all configurations. We evaluated several hyperparameter configurations, testing a sequential setting ($B = 1, K = 1$) and batch

settings with $B = 100$ combined with top sample values of $K \in \{10, 100\}$. For all tested configurations, we varied the rank from 3 to 5.

The results for all five benchmarks are presented in Figure 9 and Table 6. From these results, we can draw several conclusions to guide our selection.

First, regarding the batch size $B$, a larger value of $B = 100$ consistently resulted in superior performance across all tasks. A larger batch size allows the optimizer to gather more information about the objective function landscape in a single round, leading to a more effective and stable search process compared to the sequential evaluation approach of $B = 1$.

Second, concerning the number of top samples $K$, the choice of $K = 10$ was most frequently associated with the best-performing configurations. This suggests that $K = 10$ strikes an effective balance. It focuses the surrogate model's training on a sufficiently elite subset of high-performing samples from the batch, while still retaining enough diversity to avoid premature convergence, which can be a risk with a very small $K$ or inefficient with a very large $K$ for PROTES.

Third, for the TT rank, we observed that performance was often very similar across ranks 3, 4, and 5, especially once the optimal $B$ and $K$ were chosen. In several tasks, the best results were identical for all three ranks. However, considering all benchmarks, ranks 4 and 5 appeared most frequently in the top configurations. We select Rank 4 as a representative choice, as it provides a robust level of model expressiveness suitable for the complexity of these problems without being unnecessarily high, thus offering a good trade-off against the potential for overfitting.

Based on this analysis, we conclude that a configuration of batch size $B = 100$, top samples $K = 10$, and rank 4 offers the best trade-off between sample efficiency and final performance for the PROTES baseline. Therefore, we adopt this setting for all main experiments presented in Section 5.

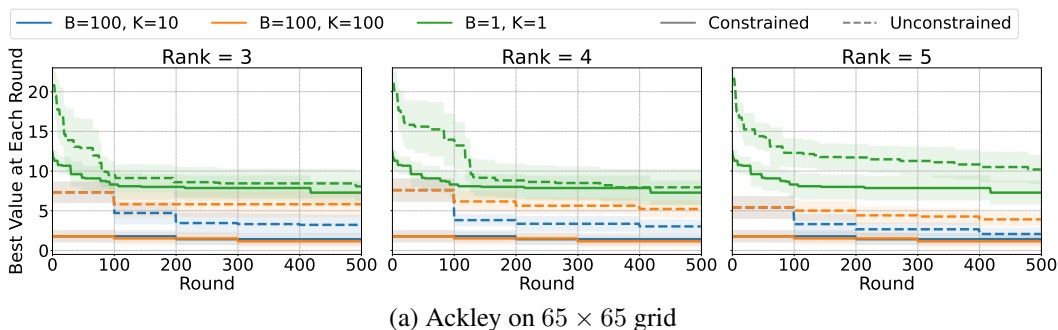

(a) Ackley on $65 \times 65$ grid

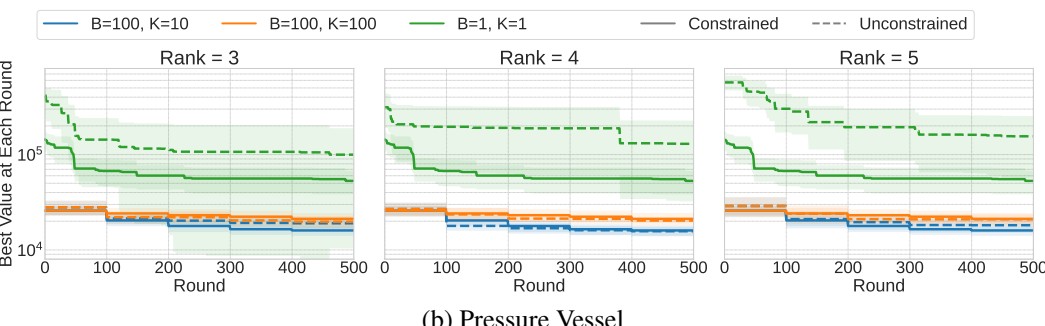

(b) Pressure Vessel

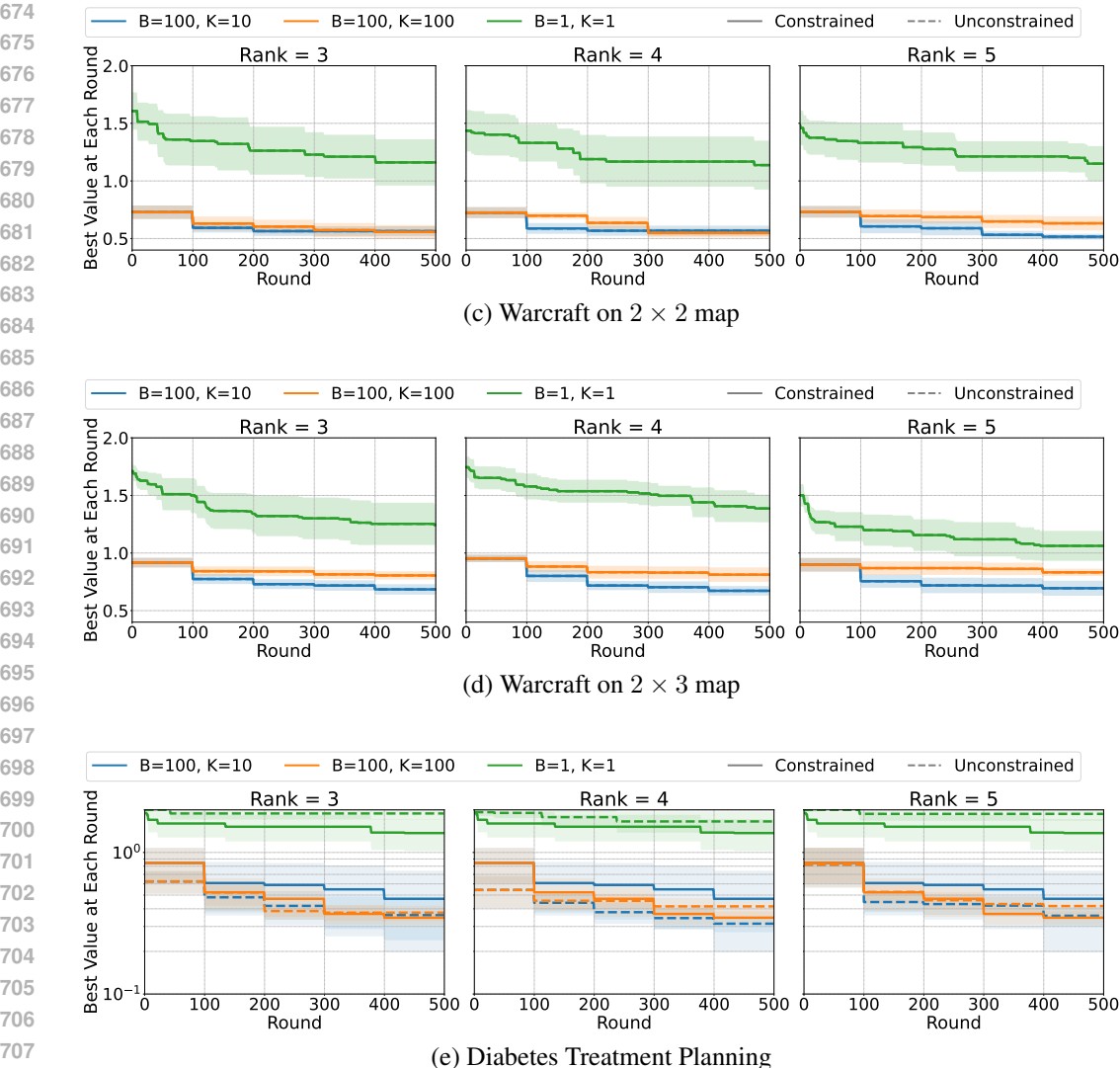

(c) Warcraft on $2 \times 2$ map

(d) Warcraft on $2 \times 3$ map

(e) Diabetes Treatment Planning

Figure 9: Ablation study for the PROTES baseline across all five benchmarks: (a) Ackley, (b) Pressure Vessel, (c) Warcraft $2 \times 2$, (d) Warcraft $2 \times 3$, and (e) Diabetes.

Table 6: Comprehensive comparison of PROTES performance across different hyperparameter configurations (Batch size B, Top samples K, and TT-Rank) on five benchmark tasks. For each configuration, we report the mean and standard deviation over ten runs for both the best feasible objective value found and the optimization round in which it first appeared.

| Objective | B | K | Rank | Best Value | Best Round |
|-----------|---|---|------|------------|------------|
| | 1 | 1 | 3 | $7.28 \pm 2.85$ | $115.10 \pm 119.14$ |
| | | | 4 | $7.28 \pm 2.85$ | $115.10 \pm 119.14$ |
| | | | 5 | $7.28 \pm 2.85$ | $115.10 \pm 119.14$ |
| Ackley | 100 | 10 | 3 | $1.42 \pm 1.44$ | $140.00 \pm 80.00$ |
| | | | 4 | $1.42 \pm 1.44$ | $140.00 \pm 80.00$ |
| | | | 5 | $1.42 \pm 1.44$ | $140.00 \pm 80.00$ |
| | | 100 | 3 | $1.15 \pm 1.44$ | $170.00 \pm 118.74$ |
| | | | 4 | $1.15 \pm 1.44$ | $170.00 \pm 118.74$ |

(Table continues on next page)

Table 6: (Continued) Summary of PROTES Ablation Study Results

| Objective | B | K | Rank | Best Value | Best Round |
|---|---|---|---|---|---|
| | | | 5 | $1.15 \pm 1.44$ | $170.00 \pm 118.74$ |
| Diabetes | 1 | 1 | 3 | $1.37 \pm 0.66$ | $61.10 \pm 130.77$ |
| | | | 4 | $1.37 \pm 0.66$ | $61.10 \pm 130.77$ |
| | | | 5 | $1.37 \pm 0.66$ | $61.10 \pm 130.77$ |
| | 100 | 10 | 3 | $0.47 \pm 0.52$ | $410.00 \pm 122.07$ |
| | | | 4 | $0.47 \pm 0.52$ | $410.00 \pm 122.07$ |
| | | | 5 | $0.47 \pm 0.52$ | $410.00 \pm 122.07$ |
| | | 100 | 3 | $0.35 \pm 0.07$ | $320.00 \pm 107.70$ |
| | | | 4 | $0.35 \pm 0.07$ | $320.00 \pm 107.70$ |
| | | | 5 | $0.35 \pm 0.07$ | $320.00 \pm 107.70$ |
| Pressure | 1 | 1 | 3 | $52982.89 \pm 27897.70$ | $211.90 \pm 170.92$ |
| | | | 4 | $52982.89 \pm 27897.70$ | $211.90 \pm 170.92$ |
| | | | 5 | $52982.89 \pm 27897.70$ | $211.90 \pm 170.92$ |
| | 100 | 10 | 3 | $16006.94 \pm 3748.15$ | $350.00 \pm 111.80$ |
| | | | 4 | $16006.94 \pm 3748.15$ | $350.00 \pm 111.80$ |
| | | | 5 | $16006.94 \pm 3748.15$ | $350.00 \pm 111.80$ |
| | | 100 | 3 | $21185.15 \pm 6004.47$ | $290.00 \pm 144.57$ |
| | | | 4 | $21185.15 \pm 6004.47$ | $290.00 \pm 144.57$ |
| | | | 5 | $21185.15 \pm 6004.47$ | $290.00 \pm 144.57$ |
| Warcraft 1 | 1 | 1 | 3 | $1.16 \pm 0.38$ | $128.50 \pm 124.38$ |
| | | | 4 | $1.14 \pm 0.40$ | $117.60 \pm 146.68$ |
| | | | 5 | $1.15 \pm 0.29$ | $180.90 \pm 198.86$ |
| | 100 | 10 | 3 | $0.56 \pm 0.08$ | $210.00 \pm 53.85$ |
| | | | 4 | $0.57 \pm 0.08$ | $190.00 \pm 53.85$ |
| | | | 5 | $0.52 \pm 0.05$ | $270.00 \pm 118.74$ |
| | | 100 | 3 | $0.56 \pm 0.11$ | $250.00 \pm 136.01$ |
| | | | 4 | $0.55 \pm 0.07$ | $300.00 \pm 109.54$ |
| | | | 5 | $0.63 \pm 0.12$ | $220.00 \pm 132.66$ |
| Warcraft 2 | 1 | 1 | 3 | $1.24 \pm 0.33$ | $207.30 \pm 153.87$ |
| | | | 4 | $1.39 \pm 0.23$ | $213.20 \pm 151.68$ |
| | | | 5 | $1.06 \pm 0.25$ | $164.30 \pm 140.55$ |
| | 100 | 10 | 3 | $0.68 \pm 0.09$ | $310.00 \pm 113.58$ |
| | | | 4 | $0.67 \pm 0.08$ | $360.00 \pm 128.06$ |
| | | | 5 | $0.69 \pm 0.12$ | $290.00 \pm 94.34$ |
| | | 100 | 3 | $0.81 \pm 0.07$ | $250.00 \pm 128.45$ |
| | | | 4 | $0.81 \pm 0.12$ | $280.00 \pm 116.62$ |
| | | | 5 | $0.83 \pm 0.06$ | $210.00 \pm 151.33$ |

# E  ADDITIONAL COMPARATIVE STUDY WITH NN+MILP

In this section, we present additional comparative experiments to further verify the effectiveness of our proposed method, CA-TD. The primary objective is to evaluate CA-TD's performance against the strong NN+MILP baseline on task domains where it has demonstrated significant success. To this end, our benchmark tasks are inspired by or directly derived from those presented in the original NN+MILP paper (Papalexopoulos et al., 2022).

A direct comparison of the original, large-scale benchmarks is challenging due to limitations in our current implementation of CA-TD. Our approach, which relies on tensor representations of the

search space, becomes memory-intensive as the problem dimensionality increases. Therefore, to ensure a computationally feasible comparison, we utilize scaled-down versions of the original tasks, with the exception of the DNA binding task where we employ the original problem specification.

### E.1 TASK DESCRIPTIONS

Within this setting, we deliberately selected a diverse set of tasks to comprehensively evaluate the robustness of our approach across varied functional landscapes. The selection spans seven new benchmark tasks across four distinct domains. These tasks are categorized into the following four classes based on the complexity of their objective functions:

- **Linear Objective Function:** We use two variants of the Generalized Assignment Problem (GAP), which represents a fundamental class of combinatorial optimization problems.
- **Quadratic Objective Function:** We use two variants of the Constrained Ising Model, which involves minimizing a quadratic function with pairwise interaction terms.
- **Complex Non-Linear Objective Function:** We use two search spaces from the Neural Architecture Search (NAS) benchmark, which features a highly complex black-box objective.
- **Biological Sequence Optimization:** We use a DNA binding affinity optimization task, representing a real-world challenge in genomics with sparse feasible regions.

A detailed description of each task domain follows.

**Generalized Assignment Problem (GAP)** GAP is a fundamental problem in combinatorial optimization, often used to model resource allocation scenarios. In this problem, we are given a set of $n$ items and a set of $m$ bins. Each item $i$ (for $i = 1, \ldots, n$) has a specific value $p_{i,j}$ and consumes a certain amount of resources (its weight $w_i$) if it is assigned to bin $j$ (for $j = 1, \ldots, m$). Each bin $j$ has a limited capacity $c_j$. The goal is to assign each item to exactly one bin in order to maximize the total value of the assignment, without violating the capacity constraints of any bin.

To formulate this mathematically, we represent an assignment as a vector $\mathbf{x} = (x_1, \ldots, x_n)$, where the element $x_i \in \{1, \ldots, m\}$ denotes the bin to which item $i$ is assigned. The total value for a given assignment $\mathbf{x}$ is the sum of the values from each assignment:

$$\text{Value}(\mathbf{x}) = \sum_{i=1}^{n} p_{i,x_i}.$$

The overall optimization problem is to find the optimal assignment $\mathbf{x}^*$ within the set of all feasible assignments $\mathcal{X}$:

$$\mathbf{x}^* = \arg\max_{\mathbf{x} \in \mathcal{X}} \text{Value}(\mathbf{x}).$$

The set of feasible assignments $\mathcal{X}$ is determined by the specific constraints of each task. It is worth noting that while the GAP instances in the original NN+MILP paper (Papalexopoulos et al., 2022) feature a more complex quadratic objective function, we specifically designed our tasks with a linear objective to ensure diversity in our task set. The details of our two linear GAP variants are as follows.

- **GAP-A (Capacity-Constrained):** This task requires assigning $n = 9$ items to $m = 3$ bins. The search space size is $3^9$. A strict equality constraint is imposed, requiring the total weight of items in each bin to exactly match its predefined capacity. Let $w_i$ be the weight of item $i$ and $c_j$ be the capacity of bin $j$. The constraint is formulated as:

$$\sum_{i=1}^{n} w_i \cdot \mathbb{I}(x_i = j) = c_j \quad \text{(for each bin } j = 1, \ldots, m),$$

where $\mathbb{I}(\cdot)$ is the indicator function. For this specific task instance, the weight of each item is set to 1, i.e., $w_i = 1$ for all $i$. This setting models resource allocation problems in which resources must be fully utilized.

- **GAP-B (Logically-Constrained):** This task involves assigning $n = 7$ items to $m = 4$ bins (search space size $4^7$). Instead of physical capacity, it features logical constraints on the number of items assigned to specific bins, formulated as:

$$\sum_{i=1}^{n} \mathbb{I}(x_i = 1) \leq 1 \quad \text{and} \quad \sum_{i=1}^{n} \mathbb{I}(x_i = 2) \leq 1.$$

This simulates operational policies or design rules where certain assignments are restricted.

**Constrained Ising Model** This task is a quadratic optimization problem over binary variables, inspired by the Ising model in physics and the ConstrainedIsing benchmark in Papalexopoulos et al. (2022). The problem involves a set of $n$ binary items, where each item can be either selected (1) or not selected (0). An interaction potential $P_{ij}$ is defined for each pair of items $(i, j)$. The objective is to choose a subset of items that minimizes the sum of potentials from all pairs of selected items.

To formulate this, we represent a selection as a binary vector $\mathbf{y} = (y_1, \ldots, y_n) \in \{0, 1\}^n$, where $y_i = 1$ if item $i$ is selected, and $y_i = 0$ otherwise. The objective function to be minimized is a quadratic form:

$$f(\mathbf{y}) = \sum_{i=1}^{n-1} \sum_{j=i+1}^{n} P_{ij} y_i y_j.$$

The overall optimization problem is to find the optimal selection $\mathbf{y}^*$ within the set of all feasible selections $\mathcal{Y}$:

$$\mathbf{y}^* = \arg\min_{\mathbf{y} \in \mathcal{Y}} f(\mathbf{y}).$$

The feasible set $\mathcal{Y}$ is determined by task-specific constraints on the selections. While the original benchmark in Papalexopoulos et al. (2022) utilizes a balancing constraint between pairs of item groups, our tasks build upon this by incorporating additional cardinality constraints to create more challenging scenarios, as detailed below.

- **Ising-A (Group-Balanced):** The task involves $n = 14$ items, partitioned into two groups of 7. The feasible set $\mathcal{Y} \subseteq \{0, 1\}^{14}$ is defined by constraints requiring that (1) the number of selected items in each group must be equal, and (2) the total number of selected items must be exactly 4. This emulates the need for balanced feature selection in matched-pair observational studies.
- **Ising-B (Complex Group-Constrained):** This task involves $n = 15$ items, partitioned into three groups of 5 (Groups A, B, and C). The feasible set $\mathcal{Y} \subseteq \{0, 1\}^{15}$ is defined by two simultaneous constraints: (1) the number of selected items in Group A must be equal to that in Group B, and (2) the number of selected items in Group C must be exactly 1. This models more complex design rules where different component groups are subject to different types of constraints.

**Neural Architecture Search (NAS)** NAS is the process of automating the design of neural networks. While the original NN+MILP paper (Papalexopoulos et al., 2022) used the NAS-Bench-101 benchmark for its case study, we employ the more recent NATS-Bench benchmark (Dong et al., 2021) for our comparative experiments to construct tasks of a more manageable scale.

Both benchmarks model a neural architecture as a directed acyclic graph (DAG) representing a computational cell, but they differ fundamentally in how the search space is defined. NAS-Bench-101 defines its vast space of 423,000 architectures by exploring the connectivity of the DAG itself, along with assigning an operation to each of its vertices. In contrast, NATS-Bench offers two distinct and more compact search spaces. Its Topological Search Space (TSS) fixes the DAG's structure and simplifies the search to selecting an operation for each edge. Its Size Search Space (SSS) keeps the topology fixed and instead searches for the optimal number of channels at different network stages.

Our tasks utilize these two search spaces from NATS-Bench. An architecture is represented by a configuration vector $\mathbf{z}$, and the objective is to find the configuration $\mathbf{z}^*$ that maximizes its pre-computed

test accuracy on the CIFAR-10 dataset (Krizhevsky et al., 2009). Test accuracy is the proportion of correct predictions over all predictions on a held-out test set. Formally, the optimization problem is:

$$\mathbf{z}^* = \arg\max_{\mathbf{z} \in \mathcal{Z}} \text{Accuracy}(\mathbf{z}),$$

where $\mathcal{Z}$ is the set of feasible architectures defined by our task-specific constraints and $\text{Accuracy}(\mathbf{z})$ represents test accuracy computed from $\mathbf{z}$.

- **TSS:** In this task, we use the TSS, where the architecture is a vector $\mathbf{z} = (z_1, \ldots, z_6)$, with each element $z_i$ being an operation on one of 6 edges. Each $z_i$ can be chosen from a set of five operations: 'none', 'skip_connect', 'avg_pool_3x3', 'nor_conv_1x1', and 'nor_conv_3x3'. The feasible set $\mathcal{Z}$ is defined by the following two constraints:

$$\sum_{i=1}^{6} \mathbb{I}(z_i = \text{'skip\_connect'}) \geq 3,$$

$$\sum_{i=1}^{6} \mathbb{I}(z_i = \text{'nor\_conv\_3x3'}) \leq 2.$$

- **SSS:** In this task, we use the SSS, where the architecture is a vector $\mathbf{z} = (z_1, \ldots, z_5)$, with each element $z_i$ being the channel count at a specific stage. Each $z_i$ can be chosen from the set of channel options $\{8, 16, \ldots, 64\}$. The feasible set $\mathcal{Z}$ is defined by the constraints:

$$\sum_{i=1}^{5} z_i \leq 160, \tag{6}$$

$$z_4 \geq z_2. \tag{7}$$

Equation 6 imposes a budget on model size, while Equation 7 represents a common design heuristic.

**DNA Binding (TfBind)** This task involves optimizing the binding affinity of a DNA sequence of length $n = 8$ to a specific transcription factor, derived from the dataset used in Papalexopoulos et al. (2022). The search space consists of $4^8$ possible sequences, where each position takes a value from the alphabet $\mathcal{A} = \{A, C, G, T\}$. We introduce a cardinality constraint on the GC-content, which is a significant structural property in genomics. Specifically, the feasible set is restricted to sequences containing at most 3 bases of type G or C. Formally, this constraint is expressed as:

$$\sum_{i=1}^{8} \mathbb{I}(x_i \in \{G, C\}) \leq 3,$$

where $\mathbb{I}(\cdot)$ is the indicator function. This constraint significantly reduces the feasible region, requiring the optimizer to navigate a sparse landscape.

### E.2 EXPERIMENTAL SETUP

For consistency in evaluation, all benchmark tasks described in Section E.1 are formulated as minimization problems. Specifically, for tasks where the original goal is to maximize a metric (e.g., total value in GAP or test accuracy in NAS), we minimize its negative value.

The overall experimental setup, including the computational environment and the number of seeds (10), follows that of the main experiments as described in Section 5.2 and Appendix C. Each optimization run is performed up to a fixed evaluation budget, which is set to 500 for all tasks in this section.

To ensure a fair comparison, we used configurations for both methods that were found to be effective in our main analysis. For our CA-TD, we used the TT format and tested with tensor ranks $r \in \{3, 4, 5\}$, based on the findings in Section 5.4. For the NN+MILP baseline, we adopted the best-performing configuration identified in the ablation study in Appendix D.3 for constrained problems

(1 initial point and 1000 training epochs). It is important to note that for this comparative study, we evaluate only the constrained variants of both methods (CA-TD and NN+MILP-c), as the focus is on performance under explicit problem constraints. Other parameters were kept identical to those in the main experiments.

### E.3   RESULTS AND DISCUSSION

The results of the comparative study are presented in Table 7, which summarizes the final performance of CA-TD (with various ranks) and NN+MILP after a fixed number of evaluations.

The results indicate that our proposed CA-TD demonstrates highly competitive performance against the NN+MILP baseline across a variety of complex combinatorial domains. Neither method consistently outperforms the other.

On the GAP-A and Ising-A tasks, both CA-TD (with optimal rank) and NN+MILP successfully identify the best objective value. However, NN+MILP achieves this in significantly fewer evaluations, thus demonstrating better sample efficiency on these specific problems. For the SSS task, NN+MILP finds a better final objective value.

Conversely, CA-TD outperforms NN+MILP on the GAP-B, Ising-B, TSS, and TfBind tasks by discovering better final solutions. This highlights the effectiveness of our approach on these particular search spaces, especially where the feasible region is sparse or the objective landscape is complex.

In conclusion, this additional study provides further evidence that CA-TD is an effective method for constrained black-box optimization. While it does not universally dominate NN+MILP, its competitive performance on these challenging tasks reinforces our central claim. The results emphasize that making the surrogate model itself aware of the feasible space is a critical and powerful strategy for developing sample-efficient constrained black-box optimizers.

Table 7: Performance comparison of CA-TD and NN+MILP on additional benchmark tasks. For each configuration, we report the mean and standard deviation over ten runs for both the best feasible objective value found and the optimization round in which it first appeared. The best-performing configuration for each task is identified based on the lowest average best value; in case of a tie, the configuration with the earlier average best round is selected. Selected best configurations are highlighted in **bold**.

| Objective | Method | Rank | Best Value | Best Round |
|-----------|--------|------|------------|------------|
| GAP A | CA-TD | 3 | -5.97 $\pm$ 0.16 | 150.50 $\pm$ 154.14 |
|  |  | 4 | -6.16 $\pm$ 0.05 | 292.30 $\pm$ 140.23 |
|  |  | 5 | -6.21 $\pm$ 0.00 | 192.50 $\pm$ 75.17 |
|  | NN+MILP | – | **-6.21 $\pm$ 0.00** | **86.40 $\pm$ 140.05** |
| GAP B | CA-TD | 3 | -4.19 $\pm$ 0.00 | 134.50 $\pm$ 74.22 |
|  |  | 4 | **-4.19 $\pm$ 0.00** | **101.80 $\pm$ 23.72** |
|  |  | 5 | -4.19 $\pm$ 0.00 | 123.40 $\pm$ 38.28 |
|  | NN+MILP | – | -4.16 $\pm$ 0.05 | 87.00 $\pm$ 109.16 |
| Ising A | CA-TD | 3 | -7.32 $\pm$ 0.00 | 171.20 $\pm$ 35.16 |
|  |  | 4 | -7.32 $\pm$ 0.00 | 111.60 $\pm$ 30.20 |
|  |  | 5 | -7.32 $\pm$ 0.00 | 110.80 $\pm$ 33.80 |
|  | NN+MILP | – | **-7.32 $\pm$ 0.00** | **78.00 $\pm$ 70.86** |
| Ising B | CA-TD | 3 | -9.06 $\pm$ 0.66 | 357.60 $\pm$ 141.67 |
|  |  | 4 | -9.43 $\pm$ 0.00 | 232.90 $\pm$ 118.71 |
|  |  | 5 | **-9.43 $\pm$ 0.00** | **206.80 $\pm$ 82.04** |
|  | NN+MILP | – | -9.38 $\pm$ 0.15 | 30.00 $\pm$ 29.03 |
| TSS | CA-TD | 3 | **-93.84 $\pm$ 0.04** | **247.10 $\pm$ 122.54** |
|  |  | 4 | -93.83 $\pm$ 0.02 | 253.80 $\pm$ 92.68 |
|  |  | 5 | -93.84 $\pm$ 0.03 | 332.00 $\pm$ 98.28 |

Table 7: (Continued) Summary of Additional Comparative Study Results

| Objective | Method | Rank | Best Value | Best Round |
|-----------|--------|------|-----------|------------|
| | NN+MILP | – | -93.82 ± 0.08 | 200.50 ± 146.20 |
| SSS | CA-TD | 3 | -91.55 ± 0.10 | 67.10 ± 37.37 |
| | | 4 | -91.72 ± 0.30 | 104.90 ± 96.53 |
| | | 5 | -91.61 ± 0.24 | 191.70 ± 137.60 |
| | NN+MILP | – | **-91.91 ± 0.19** | **215.80 ± 124.49** |
| TfBind | CA-TD | 3 | -0.9825 ± 0.0097 | 270.40 ± 121.26 |
| | | 4 | -0.9868 ± 0.0090 | 341.80 ± 153.87 |
| | | 5 | **-0.9940 ± 0.0045** | **362.50 ± 77.71** |
| | NN+MILP | – | -0.9935 ± 0.0055 | 297.50 ± 175.74 |

## F CONSTRAINT VIOLATION DURING TRAINING

This section reports how frequently constraint violations occur during optimization for both CA-TD and PGRAD. The Figure 10 shows the cumulative averaged number of points proposed by CA-TD during SMBO optimization that were rejected due to constraint violations. We observe that the number of rejected points does not increase throughout the optimization process, indicating that although CA-TD does not theoretically guarantee feasibility, it practically tends to propose feasible points almost exclusively. The Figure 11 presents the fraction of constraint-violating samples observed during training of the surrogate model when applying PGRAD. The vertical axis represents the ratio of violated samples to the total number of sampled points. These results are obtained under the setting where no unobserved points remain. The plot demonstrates that PGRAD maintains feasibility for nearly all sampled points during training, with only a very small fraction violating the constraints. Overall, the empirical evidence shows that CA-TD rarely produces infeasible proposals in practice, and PGRAD is able to prevent constraint violations in almost all cases throughout the training process.

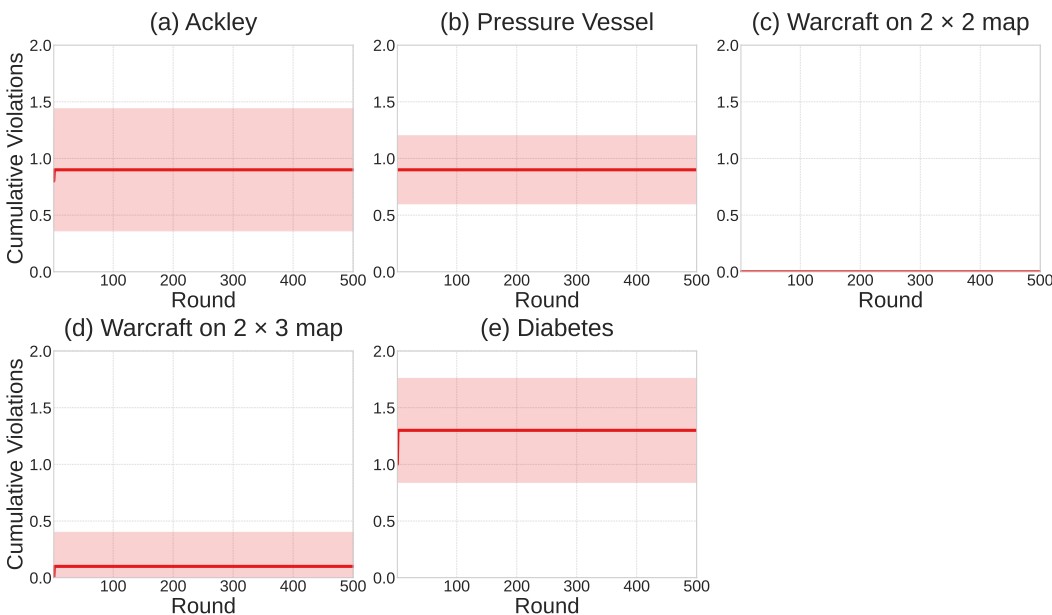

Figure 10: Averaged cumulative number of rejected points proposed by CA-TD during SMBO optimization.

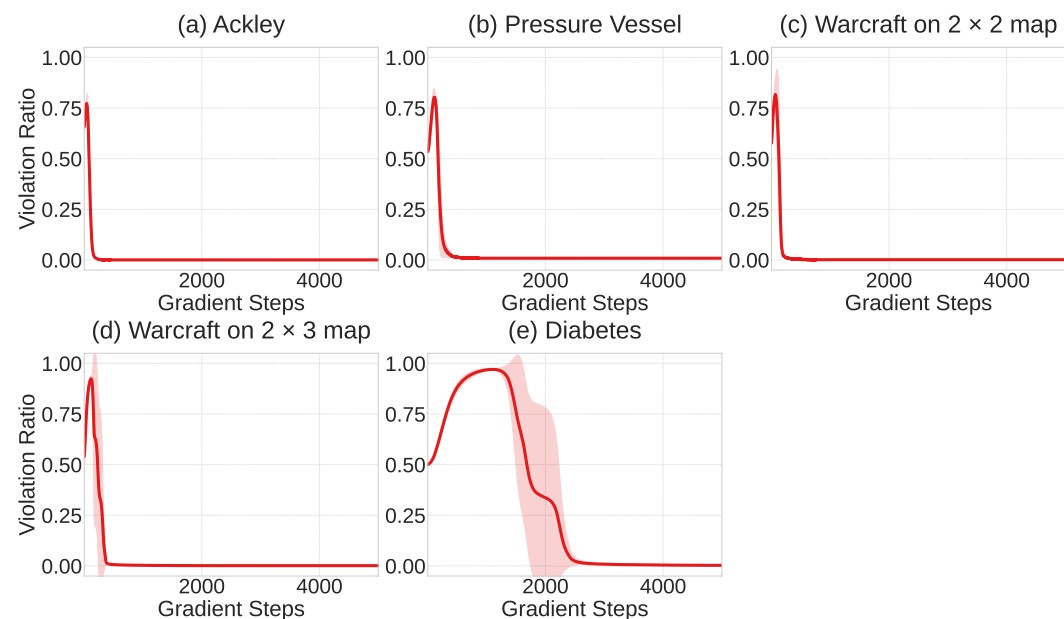

Figure 11: Fraction of constraint violations observed during training of the surrogate model in PGRAD. The vertical axis shows the ratio of violated samples to all sampled points in the gradient step.

## G CONNECTION TO POLYNOMIAL OPTIMIZATION PROBLEMS (POPS)

We clarify why the constrained surrogate learning problem becomes a Polynomial Optimization Problem (POP) by using the TT-based surrogate representation.

The surrogate tensor is represented in Tensor Train (TT) format (Equation 1). Expanding the matrix products yields

$$\hat{\mathcal{Y}}[\mathbf{x}] = \sum_{a_0,\dots,a_d} G^{(1)}_{a_0,a_1}[x_1]\, G^{(2)}_{a_1,a_2}[x_2] \cdots G^{(d)}_{a_{d-1},a_d}[x_d],$$

where $a_0,\dots,a_d$ are the indices for summing in the range 1 to $r_0,\dots,r_d$, respectively ($r_0 = r_d = 1$). Thus, $\hat{\mathcal{Y}}[\mathbf{x}]$ is a $d$-degree multivariate polynomial related to the TT core tensor parameters.

The squared error term of the learning objective in Equation 4 is also a polynomial since $\hat{\mathcal{Y}}[\mathbf{x}]$ is a polynomial in the TT parameters. Therefore, the entire objective function is a polynomial in the decision variables.

Input constraints impose $\hat{\mathcal{Y}}[\mathbf{x}] \geq \tau$ for all $\mathbf{x} \in \mathcal{X}_{\text{infeas}}$. Because $\hat{\mathcal{Y}}[\mathbf{x}]$ is a polynomial, each constraint can be written as

$$\hat{\mathcal{Y}}[\mathbf{x}] - \tau \geq 0,$$

which is a polynomial inequality in the TT parameters.

A POP is generally written as

$$\begin{aligned} \text{minimize} \quad & f(\theta) \\ \text{subject to} \quad & g_i(\theta) \geq 0, \quad i = 1,\dots,m, \end{aligned}$$

where both the objective $f$ and constraints $g_i$ are polynomials in the decision variables $\theta$.

The TT-based constrained surrogate learning problem matches this form:

- Decision variables $\theta$: all TT core tensor elements $G^{(k)}[x_k]$,
- Objective $f(\theta)$: polynomial least-squares error,
- Constraints $g_i(\theta) \geq 0$: polynomial lower-bound constraints $\hat{\mathcal{Y}}[\mathbf{x}] - \tau \geq 0$ ($m = |\mathcal{X}_{\text{infeas}}|$).

Because both the objective function and all constraints are multivariate polynomials in the TT parameters, the constrained surrogate model learning problem constitutes a POP.

## H    SCALABILITY ON HIGH-DIMENSIONAL DISCRETE SPACES

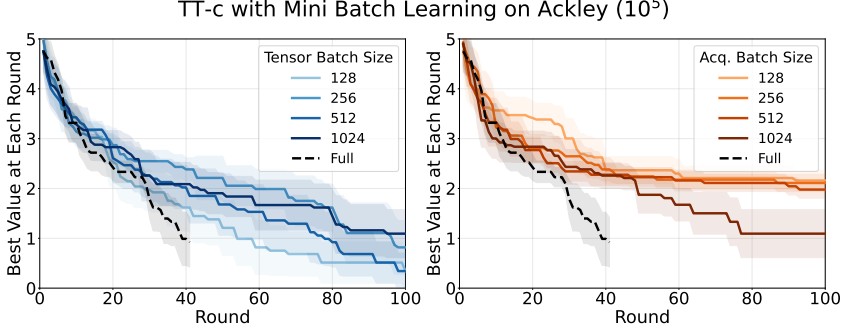

Figure 12: Ablation study on batch size for a search space of size $10^4$. (Left) **Tensor Batch Size**: Varying the training batch size for PGRAD. (Right) **Acq. Batch Size**: Varying the inference batch size for acquisition while fixing the training batch size to 128.

Figure 13: Ablation study on batch size for a search space of size $10^5$. (Left) **Tensor Batch Size**: Varying the training batch size for PGRAD. (Right) **Acq. Batch Size**: Varying the inference batch size for acquisition while fixing the training batch size to 128.

In this section, we present data on the memory usage bottleneck of tensor decomposition-based methods, which are the bottleneck of our proposed method. We then demonstrate the impact of batching tensor decomposition, the most naive solution to this bottleneck.

Standard gradient-based optimization methods in deep learning frameworks (e.g., PyTorch) generally require single-precision floating-point (Float32) formats to ensure numerical stability.

Storing dense tensors using single-precision floating-point numbers requires 0.4 GB of memory for a search space size of $10^8$, 4.0 GB for $10^9$, and 40.0 GB for $10^{10}$.

Next, we introduce a mini-batch approach to CA-TD by switching to stochastic optimization. The mini-batch method used here approximates both the loss function in Eq. 5 and the acquisition function described in Section 3.4 through sampling. For the loss computation in tensor decomposition, all previously observed points that satisfy the constraints are always included in each batch, while the remaining batch elements are randomly sampled from the constraint-violating points. For the acquisition function, mini-batches are constructed by uniformly sampling indices from the entire search space. Furthermore, the gradient descent algorithm is fixed to 200 steps, and at each step only the constraint-violating points are resampled.

This mini-batch method is expected to significantly reduce computational and memory loads by not loading the entire search space into memory.

## H.1 ABLATION STUDY ON BATCH SIZE

First, we investigate the impact of mini-batch size selection on optimization.

**Experimental Setup**    We utilize the 4-dimensional Ackley function discretized with 10 levels per dimension, resulting in a search space of size $10^4$. The feasible region is defined by the constraint $\sum_{i=1}^{d} x_i^2 \leq 3^2$. The computational environment follows the specifications described in Appendix C. The "Full" baseline (black dashed line) represents the ideal setting where both training and acquisition are performed using the full dataset (full-batch). Larger batches improve convergence toward the full-batch baseline.

**Methodology and Results**    We conduct two separate analyses by varying the batch size in $128, 256, 512, 1024$, focusing on its effect on (1) tensor training and (2) acquisition inference.

1. **Effect of Tensor Batch Size (Surrogate Training):** We vary the batch size used for the PGRAD updates while keeping the acquisition evaluation in full-batch mode. As shown in the left panels of Figure 12 and 13, reducing the training batch size substantially degrades performance: smaller batches (e.g., 128) lead to noticeably slower convergence and poorer final objective values. In contrast, larger batches behave similarly to the full-batch baseline, indicating that sufficient tensor batch size is critical for stable surrogate training.

2. **Effect of Acquisition Batch Size (Inference):** We vary the batch size used during the acquisition evaluation while fixing the tensor training batch size to 128. The right panels of Figure 12 and 13 show that changes in acquisition batch size have almost no effect on the optimization trajectory. All curves corresponding to different batch sizes closely overlap, indicating that acquisition inference is robust to batch size variation.

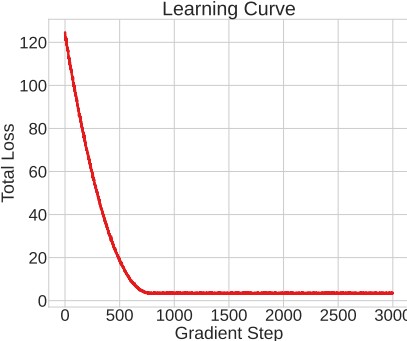

Figure 14: PGRAD learns dense $10^{10}$ Ackley function with minibatch learning with batch size 256

## H.2 APPLYING MINI-BATCHED CA-TD TO HIGH-DIMENSIONAL DISCRETE SPACE TASK

Next, we conduct experiments in a search space of size $10^{10}$ to demonstrate that the proposed mini-batch method enables optimization even when direct full-batch tensor decomposition is computationally infeasible.

With a mini-batch size of 1024, the execution time per iteration was approximately 4.43 ms, and the convergence behavior is shown in Figure 14. Figure 15 further compares CA-TD with NN+MILP on the same task. The results show that CA-TD, which leverages constraint information, achieves superior performance in the early optimization phase. In contrast, NN+MILP attains better performance in the later stages.

This performance shift is consistent with the analysis in the previous section: mini-batched CA-TD is sensitive to the batch size used for surrogate training, and in large-scale problems that require

highly accurate tensor decompositions, approximation errors accumulate during the later optimization phase. As a result, CA-TD's effectiveness diminishes over time, whereas NN+MILP maintains stable performance throughout.

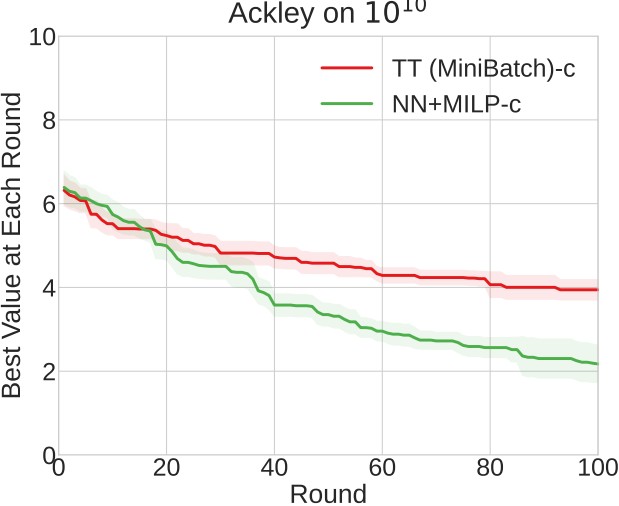

Figure 15: Optimization progress on the constrained 10-dimensional Ackley problem ($10^{10}$ search space). CA-TD (TT-c) with the mini-batch strategy demonstrates scalability and effective optimization performance compared to the NN+MILP-c baseline.

## I    ADDITIONAL FIGURE

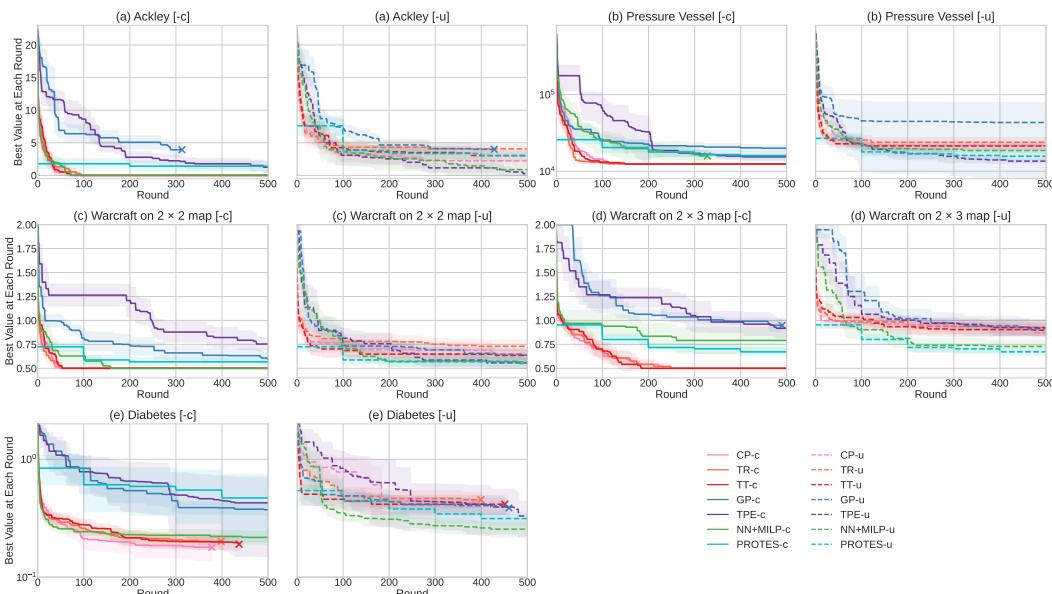

Figure 16: For ease of viewing, the same figure as Figure 3 is shown separated into -c and -u.

