# OpenReview forum: "Constraint-Aware Discrete Black-Box Optimization Using Tensor Decomposition"
_ICLR.cc/2026/Conference — Submitted to ICLR 2026_

### Official Review · Reviewer_pt1C · 2025-10-20

**Soundness:** 2
**Presentation:** 3
**Contribution:** 2
**Rating:** 4
**Confidence:** 4

**Summary:**

This paper presents CA-TD, a constraint-aware approach to discrete black-box optimization that explicitly incorporates feasibility information into a tensor-decomposition-based surrogate model. By leveraging structured tensor representations and penalty-based constraint handling, the method demonstrates competitive sample efficiency across a range of discretized benchmark problems, including synthetic functions and combinatorial optimization tasks.

**Strengths:**

1. It introduces CA-TD, a constraint-aware tensor decomposition framework that explicitly embeds feasibility constraints into the surrogate model.
2. The paper is well-organized and clearly written.
3. The authors provide transparent discussion of practical considerations and thoughtful directions for future work.

**Weaknesses:**

1. Lack novelty: The paper’s approach to constraints—assigning infeasible points poor objective value and including a penalty term in the loss—is essentially the standard practice in many constrained black-box optimization settings. In fact, several commonly used benchmark problems are constructed exactly this way.
2. The empirical evaluation is narrow: the paper only considers four small-scale benchmark problems and omits comparisons with several established baselines for constrained discrete optimization.
3. The penalty-based constraint handling (assigning poor values to infeasible points and adding a loss penalty) is a standard technique used widely in both continuous and discrete optimization. Why develop and evaluate the method only in the discrete setting?
4. It’s unclear whether the points proposed by CA-TD during optimization are guaranteed to be feasible. If not, what fraction of the evaluated points actually satisfy the constraints?
5. As noted in Section 3.4, CA-TD trains M separate surrogate models. How does this affect training time compared to baselines?
6. The paper does not release code.
7. There are minor typos. For example, on line 72, “known a priori” should be “known as a prior”.

**Questions:**

See Weaknesses.

---

> ### Author Response · Authors · 2025-11-26
>
> We deeply appreciate your careful reading of our paper, your evaluation of both its theoretical and experimental validity, and the many constructive suggestions you have provided. In the revised version, we have made the following improvements:
>
> ## Weakness 1. Novelty of Constraint-Aware TD Surrogate Training
>
> As you mentioned, using penalty terms in model training is not itself a novel idea. The contribution of our work lies in formulating the training of a tensor decomposition (TD)–based surrogate model with input constraints as a Polynomial Optimization Problem (POP), and in introducing constraint-aware optimization methods directly into the surrogate-model training.
>
> We emphasize that we are not adding penalty terms to the constrained black-box optimization problem itself (i.e., the acquisition function), but rather to the loss function of the surrogate model. The standard approach in constrained black-box optimization—including NN+MILP and other related continuous constrained BBO methods—is to impose constraints directly on the acquisition function. However, this approach creates a discrepancy between the optimization of the surrogate and the optimization of the outer black-box process. Our method is novel in that it addresses this separation by enforcing constraint-awareness during surrogate learning itself.
>
> ## Weakness 2.  Challenges and Promise of Extending to Continuous or Mixed
>
> Due to space limitations, the main paper focuses on commonly used benchmarks and practical application tasks. Nonetheless, for a more thorough comparison with NN+MILP, we performed an additional ablation study using the exact evaluation datasets from the NN+MILP paper. The results are included in Appendix E.
>
> ## Weakness 3. Challenges and Promise of Extending to Continuous or Mixed Domains
>
> Thank you for your insightful suggestion. As also related to Weakness 1, extending our method beyond penalty-based constraints would require redesigning not only the penalty scheme but the entire SMBO pipeline. Applying our approach naïvely to continuous black-box optimization introduces several challenges. For example, continuous BBO methods often use smooth surrogate models such as Gaussian Processes, and introducing constraints may create discontinuities that need careful handling. Furthermore, acquisition functions and their maximization procedures must be redesigned for such discontinuous surrogate models.
>
> That said, extending our framework to continuous or mixed discrete–continuous spaces is indeed a promising direction. We have added a discussion of this potential in the Conclusion section.
>
> $$ Weakness 4. Frequency of Constraint Violations in CA-TD
>
> To address your question about how often CA-TD actually violates constraints, we added a new figure in Appendix F. While CA-TD does not provide a theoretical guarantee of feasibility for proposed points, our measurements across all benchmarks show that the cumulative number of rejected (infeasible) points does not increase as optimization progresses. This indicates that CA-TD, in practice, proposes feasible points almost all the time.
>
> ## Weakness 5. Computational Cost and Parallel Scalability
>
> Because a tensor decomposition must be performed for each surrogate, the computational cost increases linearly with the ensemble size $M$. However, under our experimental setup with a 1-hour timeout, our method consistently matched or exceeded the performance of baseline methods, demonstrating that the current implementation is already practical.
>
> Moreover, the tensor decompositions are independent and can be parallelized when computational resources permit. On a single machine, however, the matrix operations involved in tensor decomposition are themselves highly parallelized and memory-intensive, so parallelization must consider trade-offs with memory consumption and per-decomposition parallelism. Scaling to multi-node environments is part of our future work.
>
> ## Weakness 6. Code Availability
>
> Regarding code availability, all source code is included in the supplemental ZIP file. We also plan to release a public GitHub repository, and the GitHub link will be properly included in the camera-ready version.
>
> ## Weakness 7.  Corrections
>
> Thank you for pointing this out. We have corrected the issue.

---

### Official Review · Reviewer_N8jF · 2025-10-29

**Soundness:** 2
**Presentation:** 2
**Contribution:** 2
**Rating:** 2
**Confidence:** 3

**Summary:**

This paper proposes Constraint-Aware Tensor Decomposition (CA-TD) for discrete black-box optimization (BBO) under known input constraints. The method aims to integrate constraint information directly into the surrogate model rather than only in the acquisition function. The surrogate model is constructed using Tensor-Train (TT) decomposition and trained either by (i) solving a constrained polynomial optimization problem (HSDP) or (ii) a Penalty with Gradient-based optimization (PGRAD).
Experiments on small-scale benchmarks (Ackley, Warcraft, and Diabetes) illustrate that CA-TD achieves faster convergence and better sample efficiency than several baselines such as GP, TPE, PROTES, and NN+MILP.

**Strengths:**

- This paper introduces constraint integration into surrogate training, which is a promising direction to tackle constrained black-box optimization problems.
- The proposed approach, CA-TD, achieves faster convergence and better performance than other baselines in a wide range of small tasks.

**Weaknesses:**

- The assumption that infeasible inputs yield objective values $≥ \tau$ (Eq. 4, L183–188), where $\tau$ is the maximum objective value observed among feasible inputs, might be problematic. This treats infeasibility as high-cost supervision, which is often incorrect and misleading. The surrogate thus learns from false target values, biasing it toward suboptimal or distorted landscapes. This issue could even be aggravated when the penalty coefficient λ is large (as shown in Appendix D.1) or the optimization operates in a higher-dimensional space. Therefore, I think this assumption undermines the validity of the proposed approach.
- All experiments are conducted on extremely small discrete domains (e.g., Ackley functions on 3×3 or 5×5 grids, and the Diabetes dataset with only $5^8$). These settings fall far short of realistic discrete or combinatorial optimization scenarios, and thus fail to demonstrate genuine scalability. Consequently, the paper’s claims regarding “sample efficiency” and “scalability” are not substantiated by the presented evidence.
- The presentation quality requires improvement; for instance, Table 2 is excessively large and extends beyond the page layout.

**Questions:**

- It is unclear whether $\tau$ and $y^*$ (line 231) refer to the same quantity or represent distinct variables. Please clarify their relationship to avoid confusion.
- The symbol T in Table 1 is not defined. It should be explicitly explained in the table caption to ensure clarity and self-containment.
- Minor points: The GitHub link in line 89 is empty.

Please also address these points mentioned in the Weaknesses section.

---

> ### Author Response · Authors · 2025-11-26
>
> We sincerely appreciate Reviewer N8jF for the thoughtful and constructive feedback. Below, we address each comment and provide additional clarification and supporting results where needed. We hope the responses satisfactorily resolve the concerns.
>
> ## Weakness 1.  Penalty Choice and Behavior in Discrete TD-Based Surrogates
>
> Setting the penalty for violated constraints to values worse than the currently observed feasible worst value ($\tau$) indeed risks distorting the predicted feasible region in continuous surrogate models such as Gaussian Processes (GPs). In contrast, our method operates in a discrete domain and relies on tensor completion under a low-rank assumption. This structure makes the surrogate model less susceptible to the excessive smoothing that can occur with GPs. Moreover, because we consider hard constraints that must be strictly satisfied throughout the entire optimization process—from early stages to termination—treating infeasibility as a high-cost supervisory signal is a natural and problem-aligned approach.
> That said, if our method is extended to soft constraints (i.e., weighted or penalty-based constraints), the determination of $\tau$ would indeed need to be handled more carefully. We acknowledge this and will consider it in future extensions.
>
> ## Weakness 2. Benchmark Size, Scalability Claims, and Sample Efficiency
>
> It is true that our benchmarks are small compared to extremely large-scale combinatorial optimization problems. However, relative to the smallest benchmark used in NN+MILP—TfBind ($4^8$)—our Diabetes task ($5^8$) is substantially larger. Thus, we emphasize that our experiments do not rely solely on artificially small toy problems but include benchmarks at least as large as those used in prior work.
>
> Furthermore, our claim regarding scalability is specifically about the practicality of PGRAD compared to HSDP, the exact solver whose computational cost is prohibitively large for real-world use. We do not intend to claim scalability relative to all existing optimization methods. In this paper, sample efficiency refers to “how well an optimization method finds good solutions under a fixed evaluation budget for the objective function.” In our experiments, with identical evaluation budgets, our method consistently achieves better Best Value than the baselines. This directly demonstrates improved sample efficiency due to incorporating constraints into the surrogate model.
>
> Regarding scalability, while handling sparse tensors and very high-order tensors is positioned as future work, we note that mini-batch tensor decomposition is a well-known practical technique for addressing high-dimensional scalability. This allows further extension to even larger search spaces, albeit with a trade-off: mini-batch approximation introduces noise. We have added an empirical study of this trade-off in Appendix H.
>
> ## Weakness 3.  Improvement of Table Layout
>
> Thank you for pointing this out. I have corrected the layout for now. I plan to reflect the corrections in the finalized version and rearrange the overall layout again.
>
> ## Question 1. Distinction Between $y^*$ and $\tau$
>
> We apologize for the insufficient explanation. These are indeed distinct variables:
>
> - $y^*$: the best (minimum) feasible objective value observed so far. It is used as the reference point for improvement in the acquisition function (EI).
>
> - $\tau$: the worst (maximum) feasible objective value observed so far. It is used as the penalty threshold during PGRAD training.
>
> Both values are dynamically updated as optimization proceeds.
> To avoid confusion, we revised the main text so that the role and update rule of $\tau$ are clearly distinguished from those of $y^*$.
>
> ## Question 2. Meaning of “T”
>
> Thank you for the correction. You are right—“T” denotes the total number of optimization rounds. We clarified this explicitly in both the table caption and the main text.
>
> ## Question 3. GitHub Link in Blind Review
>
> The GitHub link is intentionally left blank due to blind review requirements. The source code corresponding to the repository is included in the supplemental ZIP file. In the camera-ready version, we will provide the correct GitHub link.

---

### Official Review · Reviewer_ReeW · 2025-10-30

**Soundness:** 3
**Presentation:** 2
**Contribution:** 3
**Rating:** 4
**Confidence:** 3

**Summary:**

This paper proposes a method called Constraint-Aware Tensor Decomposition (CA-TD) for solving discrete optimization problems with constraints.
The idea is to make the surrogate model itself aware of constraints during training, instead of ignoring infeasible points until later.
The authors use a Tensor-Train (TT) model to represent the function efficiently and add a penalty loss that pushes the model to predict bad (high) values for infeasible points.
They also discuss two versions:

- HSDP, which gives theoretical guarantees using semidefinite programming but is slow.

- PGRAD, which uses gradient descent with a soft penalty and is fast and practical.

The experiments show that CA-TD works better than standard methods like GP, TPE, and PROTES on several benchmark problems.

**Strengths:**

- Clear motivation: infeasible regions waste time in discrete optimization, and the method directly handles this.

- Novel idea: combines tensor decomposition with constraint-aware learning.

- PGRAD is simple and can be trained with standard gradient-based tools.

**Weaknesses:**

- The PGRAD version has no formal theoretical guarantee of constraint satisfaction — it works empirically but not proven mathematically.

- The mathematical explanation of POP, SDP, and PGRAD is quite dense and difficult to follow for readers without background in polynomial optimization or semidefinite programming.

- The paper tests CA-TD on five benchmarks and four baselines, which is good but still limited for ICLR standards. Larger or more diverse benchmarks (e.g., higher-dimensional discrete tasks) would make the claims stronger.

**Questions:**

1- How often does PGRAD violate constraints in practice? Can you report the fraction or percentage of infeasible predictions during training?

2- How sensitive is the approach to the choice of the penalty parameter
𝜆 ? Did you tune it separately for each benchmark, or was it fixed across experiments?

3- Is the feasibility threshold 𝜏 updated dynamically based on the current feasible region, or is it fixed once per task?

4- Could the proposed CA-TD framework be extended to handle mixed discrete-continuous optimization problems in the future?

---

> ### Author Response · Authors · 2025-11-26
>
> We deeply appreciate your careful reading of our paper, your evaluation of both its theoretical and experimental aspects, and the many constructive suggestions you provided. In the revised version, we have made the following improvements:
>
> ## Weakness 1 / Question 1. Frequency of Constraint Violations in PGRAD
>
> Thank you for the insightful comment. To address your question, “How frequently does PGRAD violate the constraints in practice?”, we added a new figure to Appendix F reporting the ratio (percentage) of infeasible predictions generated during training.
>
> For PGRAD, we measured the proportion of sampled points during surrogate model training that violated the constraints. Although the number of violations does not decrease monotonically—because gradient-based learning is influenced by more than just the penalty term—we found that in the final stages the violation rate becomes extremely small, with virtually all samples satisfying the constraints. This demonstrates that, despite lacking explicit theoretical feasibility guarantees, PGRAD maintains feasible solutions with high frequency in practice.
>
> ## Weakness 2. Clarifying POP Derivation and Correspondence
>
> Thank you for pointing this out. We revised the text to improve clarity, and we added a detailed derivation in Appendix G explaining how the tensor-based formulation used in our paper corresponds to general POPs. This should make the connection accessible even to readers without prior background in the area.
>
> ## Weakness 3. Additional Comparison with NN+MILP
>
> Due to space limitations, the main paper focuses on commonly used benchmarks and practically relevant tasks. However, to provide a more thorough comparison with NN+MILP, we conducted an ablation study using the evaluation datasets employed in the NN+MILP paper. The results are included in Appendix E.
>
> ## Question 2. Sensitivity to the Penalty Parameter
>
> As shown in Appendix D.1, our method is robust with respect to the penalty coefficient $\lambda$. In all major experiments, we used a fixed value ($\lambda = 1.0$) without any task-specific tuning.
>
> ## Question 3. Updating the Threshold
>
> As optimization proceeds and new data are collected, the threshold $\tau$ (the worst observed feasible value) is dynamically updated. We have revised the main text to explicitly state this update mechanism.
>
> ## Question 4. Extension to Mixed Continuous–Discrete Domains
>
> Regarding the extension to mixed discrete–continuous optimization: yes, we believe this is both feasible and promising. Prior work such as TTOpt (Sozykin et al., 2022) has proposed tensor-decomposition–based continuous optimization using adaptive discretization. Integrating such techniques into our framework would allow the method to handle mixed discrete–continuous domains. We now mention this direction explicitly in the Conclusion section.

---

### Official Review · Reviewer_Xf3J · 2025-10-30

**Soundness:** 2
**Presentation:** 3
**Contribution:** 2
**Rating:** 2
**Confidence:** 3

**Summary:**

This paper addresses the problem of incorporating known input constraints into surrogate models for discrete black-box optimization (BBO). The authors propose CA-TD, a method that integrates feasibility information directly into the training of a low-rank tensor decomposition surrogate model, specifically using the Tensor-Train (TT) format. The core idea is to formulate the surrogate learning as a constrained Polynomial Optimization Problem (POP), solved either via (computationally heavy) hierarchical SDP relaxation (HSDP) or, more scalably, a gradient-based method with a penalty term (PGRAD). The paper demonstrates that this constraint-aware surrogate modeling improves sample efficiency over methods that handle constraints only at the acquisition function stage for a particular number of problem with small search space.

**Strengths:**

- The core idea of "constraint-aware" surrogate modeling is rather novel.
Previous work (e.g., NN+MILP) handle constraints in the acquisition step. But method PROTES, with which the comparison is made, also constructs ONE tensor during sampling, combining both the data tensor and the constraint tensor into a single object. In this sense, something similar has already been encountered.

- The experimental section is a major strength. The authors take several benchmarks, including a diverse set of tasks (Ackley, Pressure Vessel, Warcraft, Diabetes) that test the method under various conditions. The comparison is extensive, pitting CA-TD against relevant baselines (GP, TPE, NN+MILP, PROTES) in both constrained and unconstrained variants.

- The paper is generally well-written, and the methodology is clearly explained.

**Weaknesses:**

- The main weakness is  in scalability to high dimensions (The Elephant in the Room): The paper correctly identifies the memory bottleneck of dense tensor representations as the primary limitation of the approach. While PGRAD improves computational scalability over HSDP, the fundamental issue of dimensionality (the "curse of dimensionality") remains largely unaddressed. The method is demonstrated on search spaces with a moderate number of dimensions (e.g., up to 8 for Diabetes). A more substantive discussion or preliminary experiments on potential pathways to mitigate this would significantly strengthen the paper.

- The additional comparative study in Appendix E is valuable but highlights a concern. The authors had to use "scaled-down versions" of the original NN+MILP tasks to make the comparison feasible. This inevitably raises the question: Is CA-TD's competitive performance a result of its intrinsic merit, or is it simply better suited to these smaller-scale problems? It would be interesting to see a direct comparison on at least one of the larger-scale problems from the NN+MILP paper, even if it requires significant engineering effort or approximation.

- While the POP formulation is elegant, the paper lacks a theoretical analysis of the PGRAD relaxation. For instance, under what conditions does the penalty method recover the solution of the constrained problem? A discussion on the approximation guarantees or the impact of the relaxation on the surrogate's behavior in the feasible vs. infeasible regions would add considerable depth.

**Questions:**

- please, see the questions in Weakness

- could the authors elaborate on concrete strategies for scaling to higher-dimensional spaces (e.g., > 20 dimensions)? For example, have you considered or experimented with sparse tensor formats, or is this a fundamental limitation of the current CA-TD framework?

- In the NN+MILP comparison (Appendix E), CA-TD wins on some tasks and loses on others. Can you provide an intuition for the problem characteristics (e.g., structure of the feasible set, smoothness of the objective) that make CA-TD particularly well-suited or ill-suited?

- The threshold $\tau$ is set to the maximum observed feasible value. Was there any exploration of more adaptive or probabilistic strategies for setting τ? A poorly chosen τ could potentially misguide the surrogate, especially in early stages.

---

> ### Author Response · Authors · 2025-11-26
>
> We sincerely appreciate Reviewer Xf3J for the thoughtful and constructive feedback. Below, we address each comment and provide additional clarification and supporting results where needed. We hope that our responses satisfactorily resolve the concerns.
>
> ## Weakness 1 / Question 2. Scalability, Memory Bottlenecks, and Mini-Batch Approximation
>
> As the reviewer correctly pointed out, our proposed method indeed incurs certain computational costs associated with tensor decomposition and acquisition function evaluation. In this work, our focus regarding computational complexity was on mitigating the NP-hardness of the optimization problem, and we did not explicitly address issues related to dimensionality. We would like to emphasize that the main contribution of our paper is not accelerating tensor decomposition itself, but rather incorporating constraints into the surrogate model within SMBO.
>
> The most significant bottleneck of tensor decomposition in terms of scalability is memory consumption. For example, in a search space of size $10^{10}$, the memory requirement is at least 40.0 GB. As noted in the "Future Work" section, if the task involves a sparse tensor, adopting sparse tensor representations could fundamentally alleviate this issue.
> In addition, there is a more direct approach: switching to stochastic optimization via our mini-batch method. We have added a discussion in Appendix H on how mini-batching can improve scalability, showing that it enables the handling of tasks with search space sizes of up to approximately $10^{10}$.
>
> However, such approximate mini-batch–based methods introduce sampling errors. In the later stages of optimization for large-scale problems—where highly accurate tensor decompositions are required—we observed degradation in performance due to the approximation error of large-scale tensor factorization. As a result, we found that while CA-TD exhibits higher sample efficiency than NN+MILP in the early phase of large-scale optimization, NN+MILP performs better in the later phase. This suggests that CA-TD is particularly beneficial in the early stages of optimization when little information is available and constraint-aware priors are especially valuable.
>
> We also investigated the effect of batch size, and additional experiments using larger batch sizes on high-performance computing resources indicate promising potential for overcoming large-scale tensor decomposition limitations.
> In summary, although scalability introduces challenges related to approximation accuracy, we demonstrate that CA-TD is still advantageous in several practical settings. Furthermore, with more accurate large-scale tensor decomposition techniques, our approach has significant future potential. We have added a detailed discussion on these scalability considerations and memory-related bottlenecks in Appendix H, along with clarifications regarding the method’s potential and limitations.
>
> ## Weakness 2 / Question 3. Task Scale and Direct Comparison on TfBind
>
> Regarding task scale, our paper includes tasks such as Diabetes, which has a search space of $5^8 \approx 3.9 \times 10^5$. This is larger than the smallest benchmark used in the NN+MILP paper (Papalexopoulos et al., 2022), namely TfBind ($4^8$). As the reviewer suggested, a direct comparison is indeed valuable. We have therefore added a direct comparison using the TfBind benchmark in Appendix E.
>
> ## Weakness 3. Trade-off Between PGRAD and HSDP
>
> PGRAD shares convergence properties similar to standard continuous optimization methods (e.g., convergence to stationary points). However, it is true that PGRAD lacks the asymptotic optimality guarantees provided by the SDP relaxation in HSDP. Thus, there is an inherent trade-off between scalability and theoretical guarantees when choosing between HSDP and PGRAD. Nevertheless, from a practical standpoint, HSDP must terminate after a finite number of iterations, and as shown in Figure 2, both methods deliver comparable performance in practice.
>
> ## Question 4. Selection of $\tau$
>
> Regarding the selection of $\tau$, we have confirmed our method to be reasonably robust. As the reviewer pointed out, a more adaptive strategy may further improve performance. However, in our setting, we have not observed phenomena where the surrogate model stagnates or deteriorates during early optimization. Indeed, in Figures 2 and 3, the “Best Value” improves consistently from the early stages.
>
> In our experiments, we assign consistently poor objective values to infeasible inputs. Therefore, if our method frequently proposed infeasible candidates, the Best Value curve should plateau early. Instead, the curve improves steadily from the beginning, indicating that the surrogate model successfully captures the feasible region and explores effective candidates.

---

### Meta-Review · Area_Chair_zWSS · 2026-01-07

**Summary:**

This paper proposes a discrete black-box optimization method using tensor decomposition. under known input constraints. The method aims to integrate constraint information directly into the surrogate model rather than only in the acquisition function. The surrogate model is constructed using Tensor-Train (TT) decomposition and trained either by (i) solving a constrained polynomial optimization problem (HSDP) or (ii) a Penalty with Gradient-based optimization (PGRAD). Experiments on small-scale benchmarks (Ackley, Warcraft, and Diabetes) illustrate that CA-TD achieves faster convergence and better sample efficiency than several baselines such as GP, TPE, PROTES, and NN+MILP.

The idea seems fairly interesting but overall, the evaluation is too narrow. This is a concern shared by all reviewers. In addition, there are concerns regarding lack of theoretical guarantees (or reasonable empirical discussion regarding the formulation), rationalization for the assumption, incremental contribution, scalability. The rebuttal however does not provide new content and is purely on clarification, largely acknowledging these limitations. I'd not base the final decision particularly on the theoretical guarantees but the rest are reasonable concerns that need to be addressed.

Taking into account both preliminary reviews and rebuttal, I'd not expect the reviewers to raise score. Based on the written feedback, I see that multiple concerns remain unaddressed and hence, I'd recommend that the authors revise the work extensively following these feedback for a future submission.

**Reviewer Concerns:**

Most concerns were not addressed. As I mentioned above, the rebuttal is largely on clarification and it basically acknowledges the raised limitation. Most importantly, the narrow scope of the experiments is a common criticism among reviewers. I also agree with this assessment.

**Reviewer Scores:**

For reasons stated above, I do not think the reviewers would upgrade the ratings in this case.

---

### Decision · Program_Chairs · 2026-01-26

Reject